# Multi-omics profiling identifies a deregulated FUS-MAP1B axis in ALS/FTD–associated UBQLN2 mutants

Laura Strohm[1] , Zehan Hu[3], Yongwon Suk[4], Alina Rühmkorf[10], Erin Sternburg[4], Vanessa Gattringer[1], Henrick Riemenschneider[1,2], Riccardo Berutti[8], Elisabeth Graf[7], Jochen H Weishaupt[6], Monika S Brill[11,12] , Angelika B Harbauer[9,10,11] , Dorothee Dormann[4,5] , Jörn Dengjel[3], Dieter Edbauer[1,2], Christian Behrends[1]

**Ubiquilin-2 (UBQLN2) is a ubiquitin-binding protein that shuttles ubiquitinated proteins to proteasomal and autophagic degradation. UBQLN2 mutations are genetically linked to the neurodegenerative disorders amyotrophic lateral sclerosis and frontotemporal dementia (ALS/FTD). However, it remains elusive how UBQLN2 mutations cause ALS/FTD. Here, we systematically examined proteomic and transcriptomic changes in patient-derived lymphoblasts and CRISPR/Cas9–engineered HeLa cells carrying ALS/ FTD UBQLN2 mutations. This analysis revealed a strong up-regulation of the microtubule-associated protein 1B (MAP1B) which was also observed in UBQLN2 knockout cells and primary rodent neurons depleted of UBQLN2, suggesting that a UBQLN2 loss-of-function mechanism is responsible for the elevated MAP1B levels. Consistent with MAP1B's role in microtubule binding, we detected an increase in total and acetylated tubulin. Furthermore, we uncovered that UBQLN2 mutations result in decreased phosphorylation of MAP1B and of the ALS/FTD–linked fused in sarcoma (FUS) protein at S439 which is critical for regulating FUS-RNA binding and MAP1B protein abundance. Together, our findings point to a deregulated UBQLN2-FUS-MAP1B axis that may link protein homeostasis, RNA metabolism, and cytoskeleton dynamics, three molecular pathomechanisms of ALS/FTD.**

## Introduction

Amyotrophic lateral sclerosis (ALS) is a neurodegenerative disorder which affects upper and lower motor neurons (Rowland & Shneider, 2001). It is the most common adult-onset motor neuron disease (Talbott et al, 2016) and is characterized by a fast progression of muscle atrophy. Patients usually die of respiratory failure within 5 yr after diagnosis (Niedermeyer et al, 2019). Some ALS patients also suffer from cognitive impairment, which can reach the extent of frontotemporal dementia (FTD). Importantly, ALS and FTD share not only clinical but also neuropathological and genetic characteristics. For instance, inclusions containing the ubiquitinated RNA-binding proteins TDP-43 (TAR DNA-binding protein 43) or FUS (Fused in Sarcoma) can be found in patients of both diseases (Arai et al, 2006; Neumann et al 2006, 2009; Kwiatkowski et al, 2009; Vance et al, 2009). Hence, ALS and FTD are often regarded as a broad neurodegenerative disease continuum (Burrell et al, 2016; Abramzon et al, 2020).

Approximately 90% of all ALS cases are sporadic, whereas 10% occur with a familial history of the disease. Genes implicated in ALS provided critical insights into the pathogenic mechanism. Most common are mutations in *C9orf72* (chromosome 9 open reading frame 72), *SOD1* (superoxide dismutase 1), *FUS* (fused in sarcoma), and *TARDBP* (encoding for TDP-43) (Roggenbuck et al, 2017). FUS and TDP-43 primarily function in mRNA processing, suggesting that disrupted RNA homeostasis is a key element in the pathogenesis of ALS (Ling et al, 2013). Accumulating evidence also points to a dysregulation of cytoskeleton dynamics in the disease as mutations in *DCTN1* (dynactin subunit 1), *PFN1* (profilin 1), and *TUBA4* (tubulin alpha 4b) are linked to familial forms of ALS (Castellanos-Montiel et al, 2020). In addition, impaired protein homeostasis has been implicated in ALS pathogenesis. Several disease-linked proteins, for example, OPTN (optineurin), SQSTM1/p62 (sequestome 1), C9orf72 and the ubiquitin ligase substrate adapter CCNF (cyclin F) have known functions in the ubiquitin-proteasome system, autophagy, or both (Ling et al, 2013; Rea et al, 2013; Wong & Holzbaur, 2014; Sellier et al, 2016; Jung et al, 2017; Lee et al, 2018). Another ALS-associated protein implicated in this functional category is UBQLN2 (ubiquilin-2) for which several missense mutations were found in ALS patients (Deng et al, 2011). *UBQLN2* is located on the X-chromosome

[1]Munich Cluster for Systems Neurology, Medical Faculty, Ludwig-Maximilians-University München, Munich, Germany   [2]German Center for Neurodegenerative Diseases Munich, Munich, Germany   [3]Department of Biology, University of Fribourg, Fribourg, Switzerland   [4]Institute for Molecular Physiology, Johannes Gutenberg-University Mainz, Mainz, Germany   [5]Institute of Molecule Biology, Mainz, Germany   [6]Division of Neurodegenerative Disorders, Department of Neurology, Medical Faculty Mannheim, Mannheim Center for Translational Neurosciences, Heidelberg University, Mannheim, Germany   [7]Institut für Humangenetik, Klinikum Rechts der Isar der Technischen Universität München, Munich, Germany   [8]Institute of Human Genetics, Helmholtz Zentrum München, Neuherberg, Germany   [9]Institute of Neuronal Cell Biology, Technical University of Munich, Munich, Germany   [10]Max Planck Institute of Neurobiology, Martinsried, Germany   [11]Munich Cluster for Systems Neurology, Munich, Germany   [12]Institute of Neuronal Cell Biology, Technische Universität München, Munich, Germany

Correspondence: christian.behrends@mail03.med.uni-muenchen.de

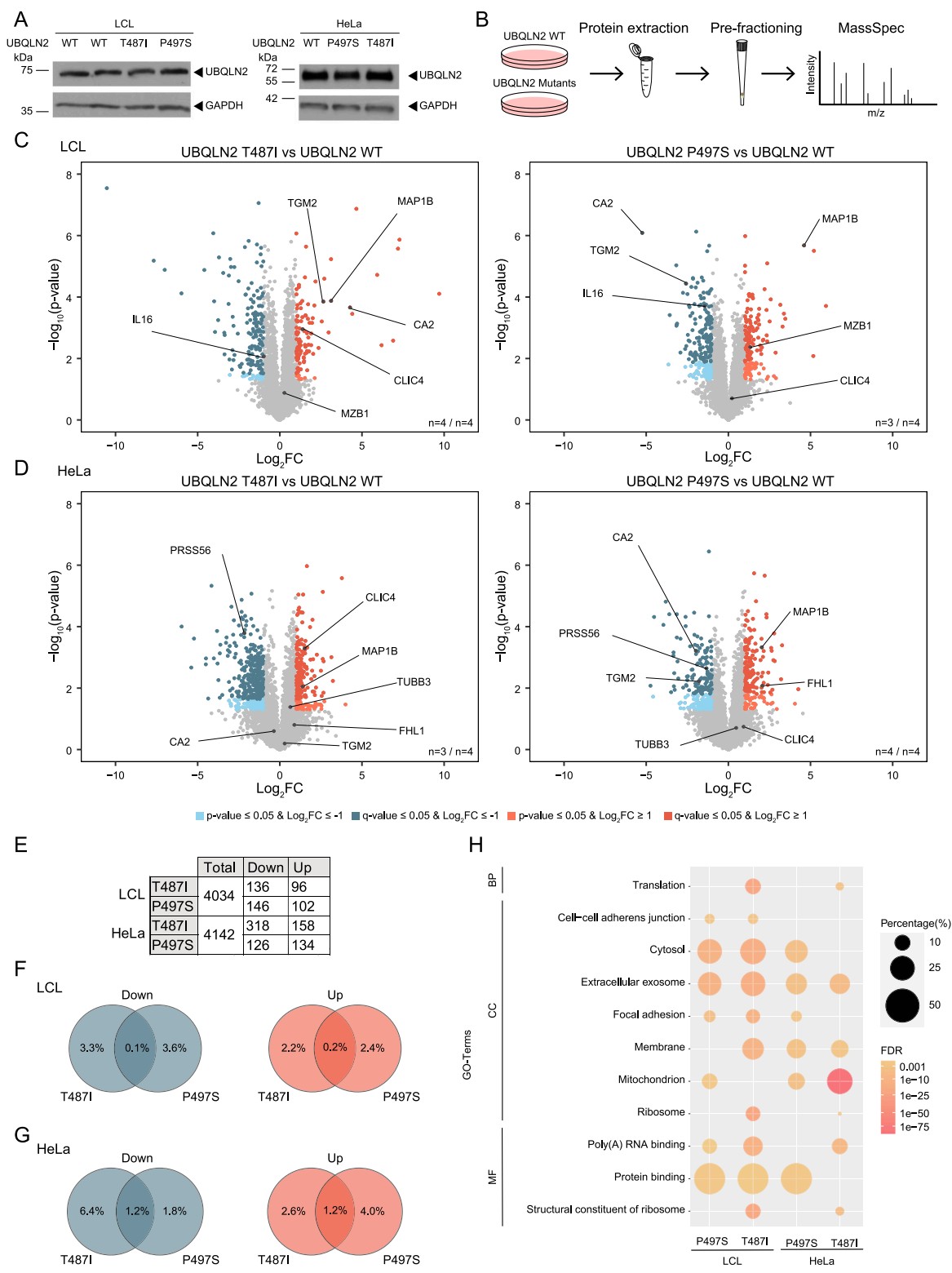

**Figure 1. Proteomics changes induced by UBQLN2 amyotrophic lateral sclerosis mutations.**
**(A)** UBQLN2 protein levels in wild-type (WT) and mutant UBQLN2 LCLs and HeLa cells. **(B)** Schematic representation of the proteomic workflow. **(C, D)** Volcano plots showing changes in relative protein abundance of patient-derived LCL (C) and engineered HeLa (D) UBQLN2 mutant cells compared with UBQLN2 WT cells. Proteins whose abundance were significantly decreased or increased in the mutant cell lines are highlighted in blue or red, respectively (log$_2$ fold change [FC] ≥ 1 or ≤ −1 $P$-value ≤ 0.05, $t$ test). Proteins passing a false discovery rate–adjusted $P$-value ($q$-value) of 0.05 are highlighted in dark blue and dark red. Candidates that are changed in multiple data sets in the same direction are highlighted. **(E)** Number of total detected proteins and those which significantly decreased or increased in abundance (log$_2$FC ≥ 1 or ≤ −1 and

and is a member of the UBQLN family. This protein family is conserved in mammals and shares a ubiquitin-associated (UBA) and a ubiquitin-like (UBL) domain and stress-inducible 1 (STI1) motifs (Wu et al, 1999; Mueller et al, 2004; Deng et al, 2011). The UBA domain enables UBQLNs to bind ubiquitinated proteins, whereas the UBL domain facilitates binding to the proteasome (Elsasser et al, 2002; Saeki et al, 2002; Dikic et al, 2009). Furthermore, UBQLNs function in autophagy, where they are involved in starvation-induced autophagy and in autophagosome maturation (N'Diaye et al, 2009; Rothenberg et al, 2010). Therefore, UBQLNs may play a role as shuttling factors in both degradation pathways. In addition to the shared UBQLN domains, UBQLN2 harbors a unique PXX domain, containing 12 tandem repeats. Interestingly, most of the known ALS-linked missense mutations are mapped in or around this region (Renaud et al, 2019). Although the exact function of the PXX motif remains unclear, there is evidence that it might modulate client specificity, proteasome binding, liquid–liquid phase separation (LLPS) of UBQLN2, and stress granule dynamics (Deng et al, 2011; Chang & Monteiro, 2015; Gilpin et al, 2015; Osaka et al, 2016; Alexander et al, 2018; Dao et al, 2019). For instance, it was shown that PXX mutant UBQLN2 fails to interact with its binding partner HSP70 and therefore is unable to shuttle protein aggregates to the proteasome (Hjerpe et al, 2016). Furthermore, experimental evidence suggests that ALS mutations in UBQLN2 impede autophagy by impairing lysosome acidification because of reduced expression of the vacuolar ATPase (Senturk et al, 2019; Wu et al, 2020). Mutations in the PXX region were also reported to promote UBQLN2 oligomerization and LLPS (Dao et al, 2019). LLPS arises through weak multivalent interactions of biomolecules, often involving RNA-binding proteins with intrinsically disordered low-complexity domains and RNAs, and can give rise to membrane-less organelles, for example, stress granules (Banani et al, 2017; Shin & Brangwynne, 2017). Taken together, impairment of protein degradation and dysregulation of LLPS and stress granule formation have been suggested to play important roles in UBQLN2-linked ALS. However, the exact mechanisms of how UBQNL2 mutations drive the disease remain elusive.

Using complementary proteomic and transcriptomic profiling, we found that ALS-linked UBQLN2 mutations (T487I and P497S) cause a robust elevation of the microtubule-associated protein MAP1B that was driven by an increase at the MAP1B transcript level and was not because of posttranslational mechanisms. Importantly, increased MAP1B protein levels were also detected in different UBQLN2 loss-of-function models, including primary neurons. Moreover, phosphoproteomic analysis of UBQLN2 mutant cells revealed a significant down-regulation of a phosphosite located in the RNA-binding zinc finger domain of the ALS/FTD–linked protein FUS. Strikingly, increased MAP1B levels in UBQLN2 KO cells could be rescued by knockdown of FUS. Moreover, phospho-ablating (S439A) and -mimicking (S439E) mutants of FUS

differentially affected FUS-RNA binding and MAP1B levels, indicating that FUS acts as an intermediator between UBQLN2 and MAP1B.

# Results

## Generation of UBQLN2 ALS mutant cells

To characterize the effect of ALS-associated mutations in UBQLN2, we examined immortalized lymphoblastoid cell lines (LCLs) derived from two ALS patients. These LCLs harbored either a T487I or a P497S mutation in UBQLN2. Both mutations were reported to cause ALS (Deng et al, 2011; Williams et al, 2012). Although the P497S mutation is located in the PXX region of UBQLN2, the T487I mutation precedes this region. As controls, LCLs from age- and gender-matched healthy individuals were used. Complementarily, we engineered HeLa cells to carry the T487I or P497S mutations in UBQLN2 using CRISPR/Cas9 technology (Fig S1). All four UBQLN2 mutant cell lines showed no difference in UBQLN2 protein levels compared with their respective UBQLN2 wild-type (WT) control cells (Fig 1A).

## Quantitative proteomics of UBQLN2 ALS mutant cells

To address whether expression of ALS-associated UBQLN2 mutant variants changes the cellular proteome, we performed global, label-free quantification-based protein abundance profiling of patient-derived LCLs and engineered HeLa cells (Fig 1B). Using this approach, we quantified over 4,000 proteins in the different cell lines carrying UBQLN2 ALS mutations (Tables S1 and S2). Filtering with a $\log_2$ fold change ($\log_2$FC) of ≥ 1 or ≤ −1 and a false discovery rate (FDR)–adjusted $P$-value (q-value) of 0.05 revealed on average 123 proteins that were significantly more and 181 proteins which were significantly less abundant compared with UBQLN2 WT cells (Fig 1C–E). Comparison of significantly enriched or reduced proteins shared between both UBQLN2 ALS mutations in the two cell types revealed a substantially smaller overlap for LCLs (Fig 1F and G). This might be explained by the heterogenous origin of these cells which are derived from patients of different age, sex, and ethnic background. Despite the differences between HeLa and LCLs, some proteins changed in the same direction in both type of cells carrying the same mutation (Fig S2). To unbiasedly analyze the data, we performed functional annotation enrichment analysis of all significantly altered proteins. Filtering of Gene Ontology (GO) terms observed in at least two of the four data sets with an FDR ≤ 0.05 revealed a number of terms known to be associated with ALS such as RNA binding and mitochondrion (Fig 1H).

---

q-value ≤ 0.05). **(F, G)** Venn diagrams of proteins from patient-derived LCL (F) and engineered HeLa (G) UBQLN2 mutant cells that were significantly (q-value ≤ 0.05) up- or down-regulated. Percentages refer to the number of total proteins found. **(H)** Gene Ontology (GO) terms enriched among significantly (q-value ≤ 0.05) altered proteins in engineered HeLa and patient-derived LCL cells carrying UBQLN2 mutations. False discovery rate–adjusted $P$-values are represented in a color gradient. Darker colors indicate higher significance levels. Percentages refer to the percentage of proteins of the input list associated to the term. BP, biological process; CC, cellular component; MF, molecular function.

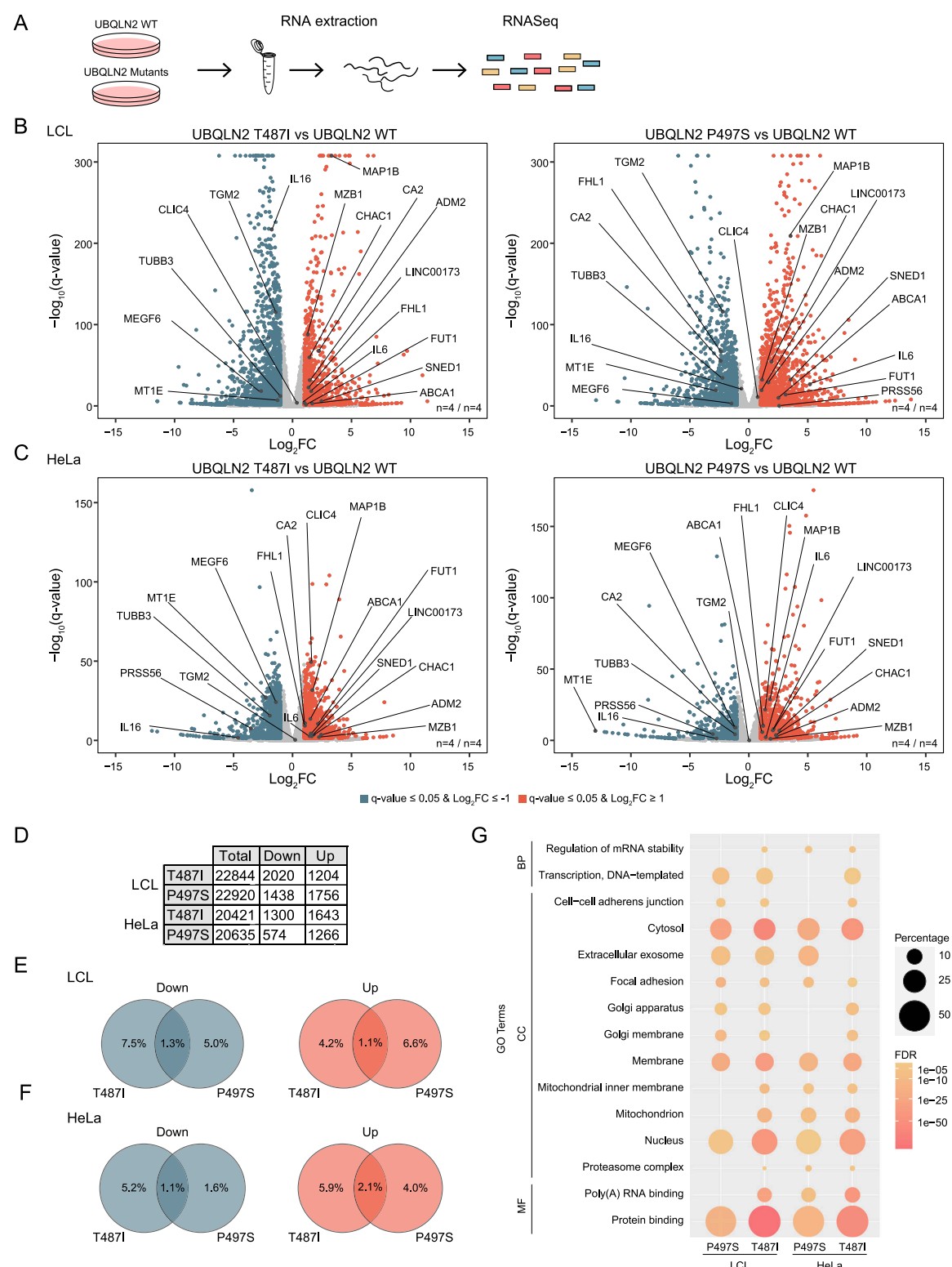

**Figure 2. Transcriptomic changes induced by UBQLN2 amyotrophic lateral sclerosis mutations.**
**(A)** Schematic representation of the transcriptomics workflow (n = 4). **(B, C)** Volcano plots showing changes in gene expression of patient-derived LCL (B) and engineered HeLa (C) UBQLN2 mutants compared with their respective UBQLN2 WT cell lines. Transcripts that are significantly decreased or increase in the mutant cell lines are highlighted in blue or red, respectively (log2FC ≥ 1 or ≤ −1; q-value ≤ 0.05, t test, false discovery rate–corrected). Candidates that are changed in multiple data sets in the same direction are highlighted. **(D)** Number of total detected transcripts and those with a significantly decreased or increased abundance (log2FC ≥ 1 or ≤

### Transcriptomics of UBQLN2 ALS mutant cells

Complementary to the proteomics approach, we subjected our cell panel to mRNA sequencing to gain broad-scale insights into changes at the mRNA level in response to the ALS mutant UBQLN2 (Fig 2A). Across the different cell lines, on average, 21,705 transcripts were quantified (Tables S3 and S4). Like, for the proteomics data, a $\log_2$FC of ≥ 1 or ≤ −1 and an FDR-adjusted $P$-value ($q$-value) of 0.05 were applied as cutoffs (Fig 2B and C). Because this data set was considerably larger, on average, 13% of all mRNA transcripts meet these criteria. Of those, on average, 1,468 and 1,333 transcripts were highly and lowly expressed, respectively, in UBQLN2 mutant cell lines (Fig 2D). Interestingly, the discrepancy in the overlap count of significantly altered transcripts between both mutants in HeLa and LCLs, as it was observed in the proteomic data, was not detected in the transcriptomic data sets (Fig 2E and F). Between 1.1% and 2.1% of the transcripts were found to be significantly regulated in the same direction for both mutations irrespective of the cell type. Also, for the transcriptomics, several genes were similarly regulated when comparing the same mutation in the two different cell types (Fig S3A). Next, we performed GO term enrichment analysis of all transcripts that were significantly up- or down-regulated in the UBQLN2 mutant cell lines. This approach unveiled a number of terms which we already observed in the GO analysis of the proteomic data (Fig 2G). These included, among others, the terms "extracellular exosome," "mitochondrial inner membrane," "mitochondrion," and "poly(A) RNA binding." In addition, we noted further ALS-related annotations. In three of four data sets, "regulation of mRNA stability" was enriched, substantiating a disturbed RNA homeostasis in mutant UBQLN2 cell lines. Furthermore, ~5% of all genes differentially regulated in cells carrying ALS-mutated UBQLN2 are associated with the Golgi, an organelle which is known to be fragmented in ALS patients (Sundaramoorthy et al, 2015). Finally, the annotations "proteasome complex" and "protein binding" could be attributed to the protein shuttling function of UBQLN2. Correlation analysis across hits that were significant in at least one data set revealed a positive correlation (Pearson's correlation coefficient ≥0.2 and $P$-value ≤ 0.05) between transcriptomics and proteomics for each mutant in LCLs but no correlation between these two mutants in either of these data sets. In contrast, for HeLa cells, we detected a positive correlation of the transcriptomics and proteomics changes in both mutant lines. However, these mutants did not correlate with each other when comparing transcriptomics or proteomics changes (Fig S3B). Because UBQLN2 plays an important role in protein degradation, we asked which protein abundance differences were because of changes at the transcriptional level. Thereto, candidates found altered in both omics data sets were compared. For HeLa between 1.3% and 8.7% of the proteomic

changes could at least partially be explained by changes in transcription, whereas this fraction reaches up to 29.4% in LCLs (Fig S3C).

### Omics analyses converge on elevated MAP1B levels in UBQLN2 ALS mutant cells

Next, we compared the top differentially expressed candidates from the two profiling approaches across the different UBQLN2 mutants and cell types. For this analysis, we only considered candidates which were changed in at least four of the eight data sets. Intriguingly, the microtubule-associated protein 1B (MAP1B) was the only candidate which was significantly altered in the same direction in all eight data sets. At the mRNA and protein level, MAP1B was up-regulated in both UBQLN2 mutants in LCLs and HeLa cells (Fig 3A). MAP1B is known to have microtubule-stabilizing properties and to regulate microtubule dynamics (Gonzalez-Billault et al, 2004). Considering that cytoskeleton alterations are a common feature in ALS pathology, we decided to focus on MAP1B. First, we sought to confirm the increased *MAP1B* mRNA levels using quantitative real time PCR (qRT-PCR). In all mutant UBQLN2 cell lines, the mRNA levels were significantly ($P$-value ≤ 0.001) higher compared with the UBQLN2 WT controls (Fig 3B). Second, we monitored the increase in MAP1B protein levels by immunoblotting. Because MAP1B is posttranslationally cleaved into a heavy and a light chain (MAP1B-LC1) (Hammarback et al, 1991) (Fig 3C), we examined both full-length MAP1B and MAP1B-LC1. Immunoblot analysis confirmed the increase in both proteoforms in mutant UBQLN2 LCLs and HeLa cells (Fig 3D). To further substantiate MAP1B up-regulation in UBQLN2 mutant cells, we performed immunofluorescence staining in LCLs. Confocal microscopy of these cells confirmed elevated levels of MAP1B in UBQLN2 mutant cells compared with their WT counterparts (Fig 3E). Notably, treatment of LCLs with either bortezomib (Btz) to block the proteasome or bafilomycin A1 (BafA1) to disrupt autophagy did not lead to elevated MAB1B levels in UBQLN2 WT cells or to a further increase in its abundance in UBQLN2-T487I mutant cells (Fig S4). Together, these findings indicate that MAP1B is a robust effector of UBQLN2 ALS mutations in HeLa and LCL cells and that the changes in MAP1B protein abundance are most likely driven by an increase at the level of MAP1B transcript.

### MAP1B up-regulation is mediated by a loss of UBQLN2 function mechanism

Presently, it is not clear whether mutations in UBQLN2 cause ALS primarily through gain- or loss-of-function mechanisms (Renaud et al, 2019). To address whether MAP1B is up-regulated in response to a loss of UBQLN2, we generated UBQLN2 KO HeLa cells. In these

---

−1 and $q$-value ≤ 0.05). **(E, F)** Venn diagrams of transcripts from patient-derived LCL (E) and engineered HeLa (F) UBQLN2 mutant cells that were up- or down-regulated. Percentages refer to the total transcripts found in the P497S mutant. **(G)** Gene Ontology (GO) terms enriched among significantly changed transcripts in engineered HeLa and patient-derived LCL cells carrying UBQLN2 mutations. False discovery rate–adjusted $P$-values are represented in a color shade. Darker colors indicate higher significance levels. Percentage numbers refer to the percentage of genes of the input list associated to the term. BP, biological process; CC, cellular component; MF, molecular function.

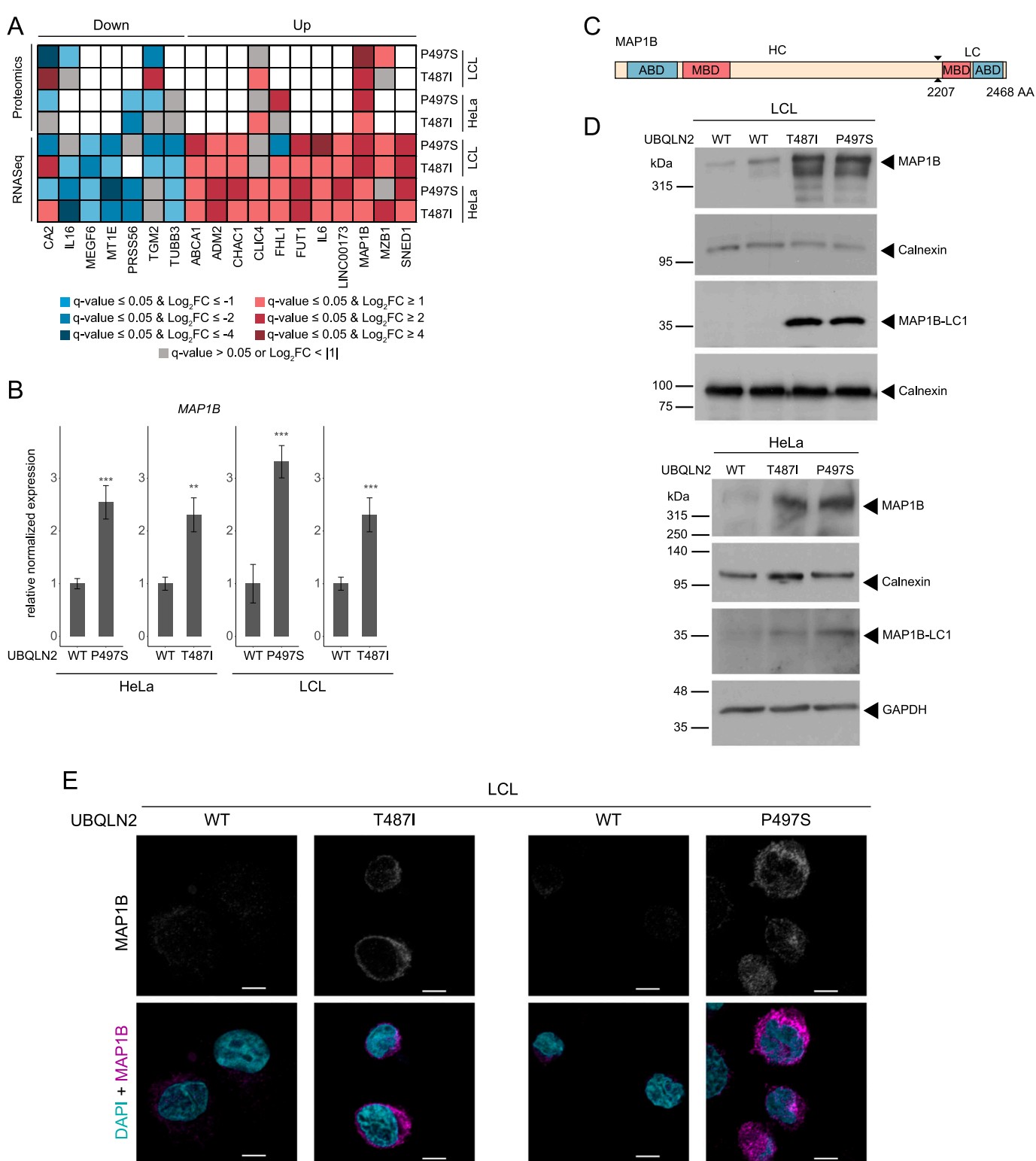

**Figure 3. UBLQLN2 mutations result in elevated MAP1B levels.**
**(A)** Common candidates of the transcriptome and proteome analyses. Only candidates which were commonly regulated in at least four of the eight omics data sets were included. Blue and red indicate significantly decreased and increased mRNA and protein abundance, respectively. **(B)** Gene expression changes of *MAP1B* in engineered HeLa and patient-derived LCL cells measured by RT-PCR. Error bars represent standard error of the mean of three biological replicates (**$P < 0.01$, ***$P < 0.001$, $t$ test). **(C)** Schematic representation of MAP1B. ABD, actin-binding domain. MBD, microtubule-binding domain. **(D)** Protein abundance changes of MAP1B full-length and light chain (MAP1B-LC1) from engineered HeLa and patient-derived LCL cells. **(E)** Representative images of patient-derived LCL cells fixed and stained with a MAP1B antibody and DAPI. Scale bar: 5 $\mu$m.

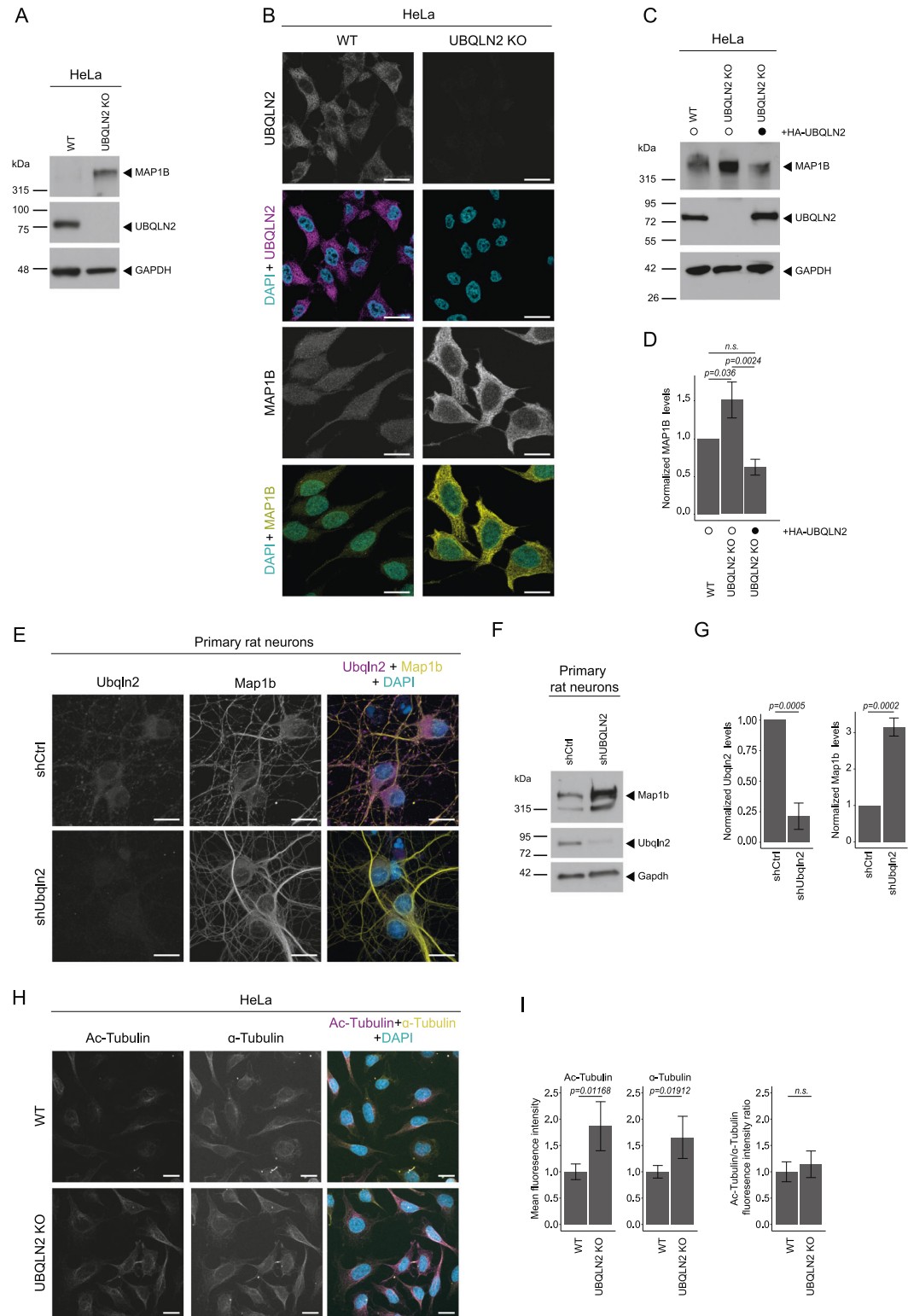

**Figure 4. Loss of UBQLN2 leads to elevated MAP1B in HeLa cells and primary neurons.**
**(A)** Immunoblot of UBQLN2 WT and KO HeLa cells. **(B)** Representative images of UBQLN2 WT and KO cells fixed and immunostained for UBQLN2 and MAP1B. Scale bar: 20 µm. **(C)** Immunoblot analysis of UBQLN2 WT and KO cells transfected with HA-UBQLN2 or left untreated (MOCK; only Lipofectamine). **(C, D)** Quantification of MAP1B protein levels from (C). Data represent mean ± SD. Statistical analysis (n = 3) of the MAP1B protein/control protein ratio was performed using one-way ANOVA followed by Tukey's post hoc test. n.s., not significant. **(E)** Ubqln2 knockdown in primary hippocampal rat neurons transduced with lentivirus expressing either *Ubqln2*-targeting shRNA

cells, we observed a strong increase of MAP1B protein levels by immunoblotting (Fig 4A) and cytochemistry (Fig 4B). Thus, UBQLN2 KO phenocopies the two UBQLN2 mutants in regard to their effects on MAP1B. Conversely, we examined whether reintroduction of UBQLN2 WT in UBQLN2 KO cells can restore normal MAP1B levels. Transient overexpression of HA-UBQLN2 in UBQLN2 KO cells indeed significantly decreased the levels of MAP1B (Fig 4C and D). As ALS is a neurodegenerative disease, we examined whether a reduction of UBQLN2 also increases MAP1B levels in neuronal cells. For this purpose, we confirmed that viral delivery of shRNA targeting *Ubqln2* in primary rat and mouse neurons decreased Ubqln2 levels (Figs 4E and F and S5A and B). Subsequently, qRT-PCR and immunoblotting showed a substantial increase of *Map1b* mRNA (Fig S5C) and MAP1B protein (Fig 4F and G) in these cells compared with their wild-type counterparts. In accordance with the microtubule-stabilizing properties of MAPs, we detected a significant elevation of total and acetylated tubulin in UBQLN2 KO cells (Fig 4H and I). However, as both proteoforms increase to a similar extend, their ratio remained unchanged (Fig 4H and I).

To further dissect the molecular consequences of elevated MAP1B, we turned to an overexpression model. HeLa cells stably expressing HA-tagged MAP1B (Fig S6A) were subjected to HA immunoprecipitation (IP) followed by mass spectrometry (Fig S6B), to identify MAP1B interacting proteins (Table S5). Parental HeLa cells were used as the negative control. Label-free quantification and statistical analysis of quadruplicate samples revealed a number of cytoskeleton-associated MAP1B interaction candidates such as TUBA4A, TUBB3, MAP1S, DYNC1H1, BUB3, CFL1, and FLNA (Fig S6C). Besides multiple proteasome subunits, the ubiquitin ligase HUWE1, the ubiquitin-binding proteins SQSTM1 and FAF2 (UBXD8), and several different types of chaperones also featured prominently among MAP1B interaction candidates, indicating that increased amounts of MAP1B might preoccupy these parts of the cell's proteostasis network.

## Phosphoproteomics of UBQLN2 ALS mutant cells

Besides acetylation, MAP1B undergoes extensive phosphorylation, which also affects its distribution and function in cells (Gonzalez-Billault et al, 2004). To examine the phosphorylation state of MAP1B, which is elevated because of the ALS mutations in UBQLN2, we performed phosphoproteomics of our LCL cell panel (Fig 5A and Table S6). In total, we identified over 40,000 phosphosites and quantified ~35,000 of those sites (Fig 5B). Among them, more than 11,000 sites were normalized to the corresponding protein amount in at least two data points (i.e., biological replicates). The overwhelming majority of these phosphosites were detected on serines (89.8%), whereas

threonines (9.7%) and tyrosines (0.5%) were found modified by far less frequently (Fig 5C). On MAP1B, we identified a total of 124 phosphosites, of which 116 were quantified, and 95 reached a localization probability higher than 0.75. Upon normalization to MAP1B protein abundance, 17 phosphosites were found to be significantly dephosphorylated ($\log_2$ ratio ≤ −1) in at least one of the two UBQLN2 ALS mutant LCLs, but only one of these MAP1B phosphosites (T1864) was significantly less phosphorylated in both mutants (Fig 5D). Notably, T1864 is located in the heavy chain of MAP1B outside of any annotated domains or motifs. The fact that not a single MAP1B phosphosite was found up-regulated indicates that surplus MAP1B protein is mostly in an hypophosphorylated state in ALS UBQLN2 mutant cells. However, further work is required to understand the downstream consequences of MAP1B dephosphorylation in response to ALS mutant UBQLN2.

Taking advantage of the phosphoproteomic data set, we searched for phosphosites in other proteins that might help to explain the up-regulation of MAP1B. Thereto, we concentrated on those sites which were commonly regulated in both UBQLN2 ALS mutants and which passed the threshold of a $\log_2$ ratio of ≥ 1 or ≤ −1 in at least one of them. With this cutoff criterion, we detected 101 phosphosites that were down-regulated and 95 that were up-regulated in response to ALS deficient UBQLN2. Unbiased functional annotation analysis of the 165 proteins which carried these phosphosites revealed a total of 19 GO terms that were significantly enriched (FDR ≤ 0.05) (Fig 5E). These included cytoskeleton-associated terms ("actin cytoskeleton," "cytoskeleton," "actin binding," and "actin filament binding") and "protein binding" and "poly(A) RNA binding." We analyzed known interactions between our phospho-candidates using the STRING database and noticed a high degree of connectivity among them (Fig 5F). Intriguingly, the cluster in which MAP1B was found included several RNA-binding proteins, of which three are genetically linked to ALS. These were heterogeneous nuclear ribonucleoprotein A1 (HNRNPA1), heterogeneous nuclear ribonucleoproteins A2/B1 (HNRNPA2B1), and fused in sarcoma (FUS).

## FUS pS439 is critical for regulating RNA binding and MAP1B abundance

From these three candidates, FUS caught our attention based on a number of reasons. First, mutations in FUS are a frequent cause for ALS (Deng et al, 2014a). Second, although the differently phosphorylated FUS residue (pS439) is not a site that is known to be mutated in ALS patients, pS439 is located in the zinc finger domain of FUS (Fig 6A) which is known to be involved

(shUbqln2) or a control shRNA (shCtrl). Five days after transduction cells were fixed and immunostained for Ubqln2 and Map1b. Maximum intensity projections of z-stack images. Scale bar: 20 μm. **(F)** Primary cortical rat neurons were transduced with lentivirus co-expressing either *Ubqln2*-targeting shRNA (shUbqln2) or a control shRNA (shCtrl) and tagRFP. Five days after transduction cells were harvested for immunoblotting. **(F, G)** Quantification of Ubqln2 and Map1b protein levels from (F). Data represent mean ± SD. Statistical analysis (n = 3) of the target protein/control protein ratio was performed using *t* test. **(H)** Representative images of UBQLN2 WT and KO HeLa cells fixed and immunostained for acetylated tubulin (Ac-Tubulin) and α-tubulin. Maximum intensity projections of z-stack images. Scale bar: 20 μm. **(I)** Quantification of Ac-Tubulin, α-tubulin, and Ac-Tubulin/α-tubulin ratio. Data represent mean gray intensity levels ± SD. Statistical analysis (n = 5, 20 cells each) was performed the using *t* test.

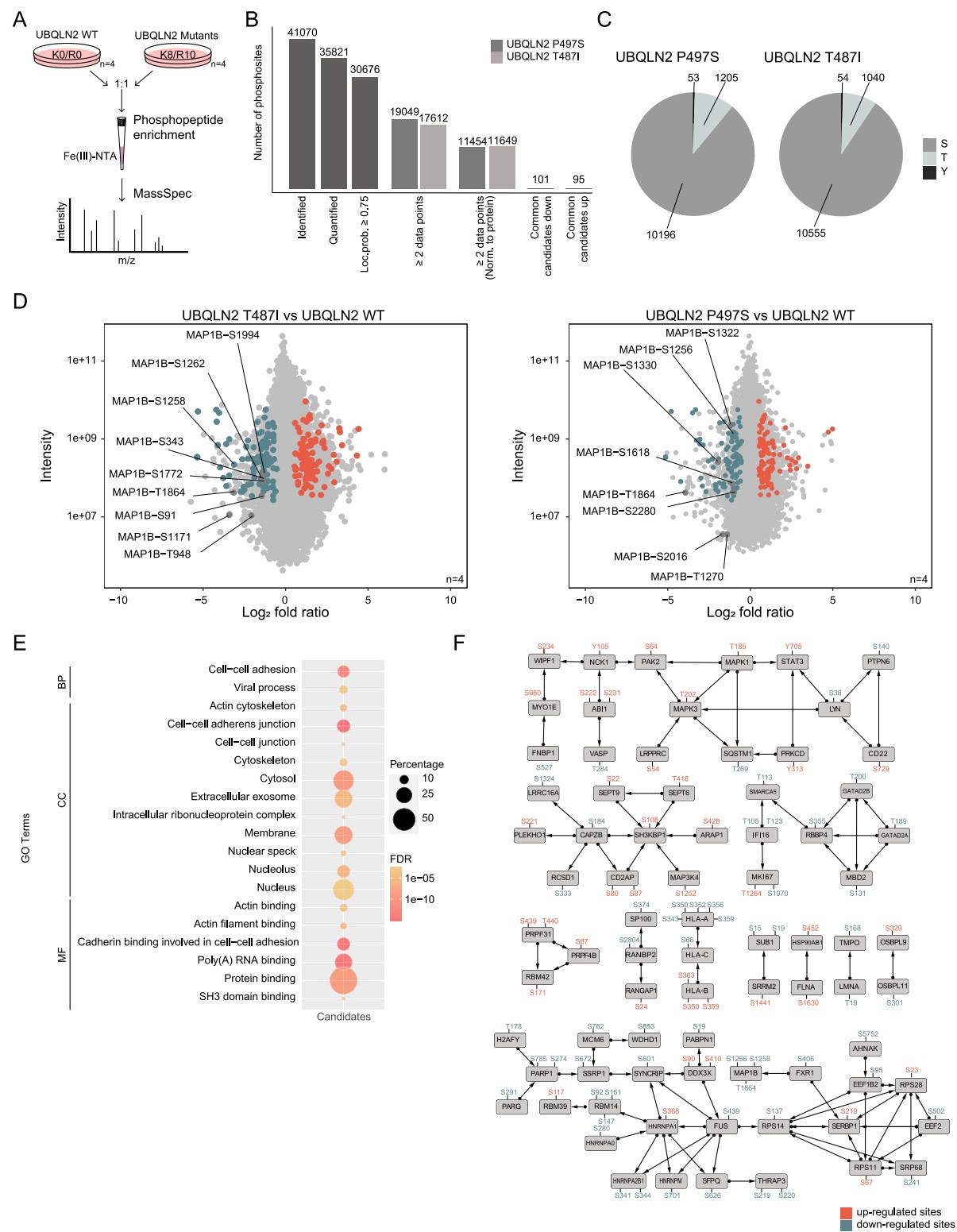

**Figure 5. Altered phosphorylation of MAP1B and FUS in response to UBQLN2 amyotrophic lateral sclerosis (ALS) mutations.**
**(A)** SILAC-based phosphoproteomics workflow in UBQLN2 WT and ALS mutant LCLs (n = 4). **(B)** Combined numbers of identified and quantified phosphosites. Data-filtering steps are indicated. **(C)** Distribution of pSer, pThr, and pTyr sites in T487I and P497S UBQLN2 ALS mutants. **(D)** Changes in cellular phosphorylation in patient-derived UBQLN2-mutant LCLs. Median log$_2$ fold ratios in phosphorylation of the identified phosphosites are presented in relation to the intensity of detection. Hypophosphorylated peptides are shown in blue and hyper-phosphorylated ones in rede. Significantly changed MAP1B phosphosites are highlighted. **(E)** Gene

in recognition and binding of RNA targets (Nguyen et al, 2011; Loughlin & Wilce, 2019; Loughlin et al, 2019). Finally, there is evidence that FUS is able to bind *MAP1B* mRNA and influences its translation (Blokhuis et al, 2016; Imperatore et al, 2020). Consistent with this notion, reanalysis of photo-activatable-ribonucleoside–enhanced cross-linking and immunoprecipitation (PAR-CLIP) data (Hoell et al, 2011) unveiled several FUS binding sites in *MAP1B* which are clustered at the 3′ UTR (Fig S7). Moreover, high confidence FUS-binding sites within *MAP1B* were also observed in two additional CLIP-seq data sets (Nakaya et al, 2013; Jutzi et al, 2020). Based on these facts, we speculate that phosphorylation of FUS at S439 may contribute to the UBQLN2-dependent regulation of MAP1B. As a first step in testing this hypothesis, we separated lysates from UBQLN2 WT and ALS-mutant LCLs on Phos-tag and SDS–PAGE gels to validate the differential phosphorylation of FUS upon UBQLN2 deficiency. Immunoblotting showed that FUS phosphorylation was indeed reduced in both UBLQN2 mutants, whereas total FUS remained unchanged (Fig 6B). To further confirm that FUS mediates the regulatory effect of UBQLN2 on MAP1B, we depleted FUS in UBQLN2 KO HeLa cells using siRNAs and examined the abundance of MAP1B. Strikingly, we noticed a significant reduction in MAP1B protein levels upon FUS knockdown (Fig 6C and D). Because key functions of FUS rely on its ability to bind RNA, we examined the role of S439 phosphorylation for FUS-RNA binding. For this purpose, we purified FUS WT, phospho-mimicking (S439E) and phospho-deficient (S439A) variants thereof and performed electrophoretic mobility shift assays (EMSA). As the RNA template, we used a synthetic *SON* pre-mRNA which consists of the stem loop and a downstream GGU motif and that was reported to simultaneously bind FUS' RRM and ZnF domain (Loughlin et al, 2019). Intriguingly, recombinant unphosphorylated FUS WT and FUS S439A bound the RNA template, whereas FUS S439E did not (Fig 6E and F), indicating that phosphorylation of S439 might be an important modulator of the RNA-binding capacity of FUS. To assess whether S439 phosphorylation of FUS has any relevance for its regulation of MAP1B, we generated FUS-knockdown (KD) HeLa cells using CRISPR/Cas9 (Fig 6G) and reconstituted these cells with HA-FUS WT or S439A (Fig 6H). Immunoblotting for MAP1B revealed that FUS S439A was not able to restrict the abundance of MAP1B compared with FUS WT (Fig 6H and I). Together, these findings support the hypothesis that the UBQLN2-mediated regulation of MAP1B is dependent on FUS and possibly involves its phosphorylation on S439 (Fig 6J).

## Discussion

Here, we present a systematic proteomic and transcriptomic profiling approach to identify changes associated with ALS-linked mutations in UBQLN2. Most of the ALS-linked genes can be assigned to a subset of molecular processes which imply possible pathogenic mechanisms of ALS. These relate primarily to RNA processing, cytoskeleton dynamics, mitochondrial function, homeostasis, and degradation. Because UBQLN2 functions mainly as a protein shuttle for ubiquitinated proteins, mutations in UBQLN2 are thought to be predominantly associated with impaired protein degradation. However, recent evidence also points to a role of UBQLN2 in modulation of RNA homeostasis (Alexander et al, 2018). In agreement with the latter, proteins and genes associated with RNA processing terms such as "regulation of mRNA stability," "poly(A) RNA binding," or "translation" were prominently represented in our proteomic and transcriptomic data sets besides additional terms such as "mitochondrion," "extracellular exosome," and Golgi-associated terms. Although morphological changes of mitochondria and the Golgi are established consequences of UBQLN2 mutations (Lin et al, 2021), the connection between UBQLN2 and extracellular exosomes has so far not been explored in great detail. However, emerging evidence links exosomes to ALS pathology. Given their role in transport of proteins and RNAs in and out of cells, exosomes are believed to be involved in the neuro-inflammation observed in ALS and might propagate protein misfolding and aggregation (Gagliardi et al, 2021). Intriguingly, proteins assigned to the term "extracellular exosome" in our analysis were not only exosome cargo but also modulators of exosome biogenesis and secretion such as Rab27A or CD9 (Ostrowski et al, 2010; Andreu & Yáñez-Mó, 2014).

Besides protein and RNA homeostasis, we provided evidence that UBQLN2 ALS mutations affect components of the cytoskeleton, in particular MAP1B. This 2,468–amino acid large protein is primarily expressed in neurons, where expression is highest during growth but persist in high plasticity areas of the adult brain (Tucker et al, 1989). As for other microtubule-associated proteins (MAPs), MAP1B was shown to bind microtubules (Noble et al, 1989; Togel et al, 1998). However, there is conflicting evidence on whether MAP1B stabilizes microtubules (Takemura et al, 1992) or acts as a factor limiting microtubule stabilization and increasing the population of dynamic microtubules (Goold et al, 1999; Gonzalez-Billault et al, 2001). The latter function is essential during neuronal development to allow axonal growth (Gonzalez-Billault et al, 2001; Tymanskyj et al, 2012). Subcellular distribution and microtubule binding properties of MAP1B are highly dependent on its phosphorylation state (Gordon-Weeks & Fischer, 2000) as it was shown for the induction of acetylated and tyrosinated microtubules (Villarroel-Campos & Gonzalez-Billault, 2014). In addition to its function in microtubule dynamics, a regulatory role in the activity and distribution of neurotransmitter receptors was also reported for MAP1B (Eriksson et al, 2010; Kim et al, 2014; Palenzuela et al, 2017).

Ontology (GO) enrichment analysis of proteins whose phosphorylation status was altered significantly in both mutants in the same direction with at least one mutant exceeding a log$_2$ fold change of ≥ 1 or ≤ −1. BP, biological process; CC, cellular component; MF, molecular function. **(F)** Protein–protein interaction (PPI) network of proteins whose phosphorylation status was significantly changed in both mutants with at least one mutant exceeding a log$_2$ ratio of ≥ 1 or ≤ −1. Significantly up- and down-regulated phosphosites are marked in red and blue, respectively.

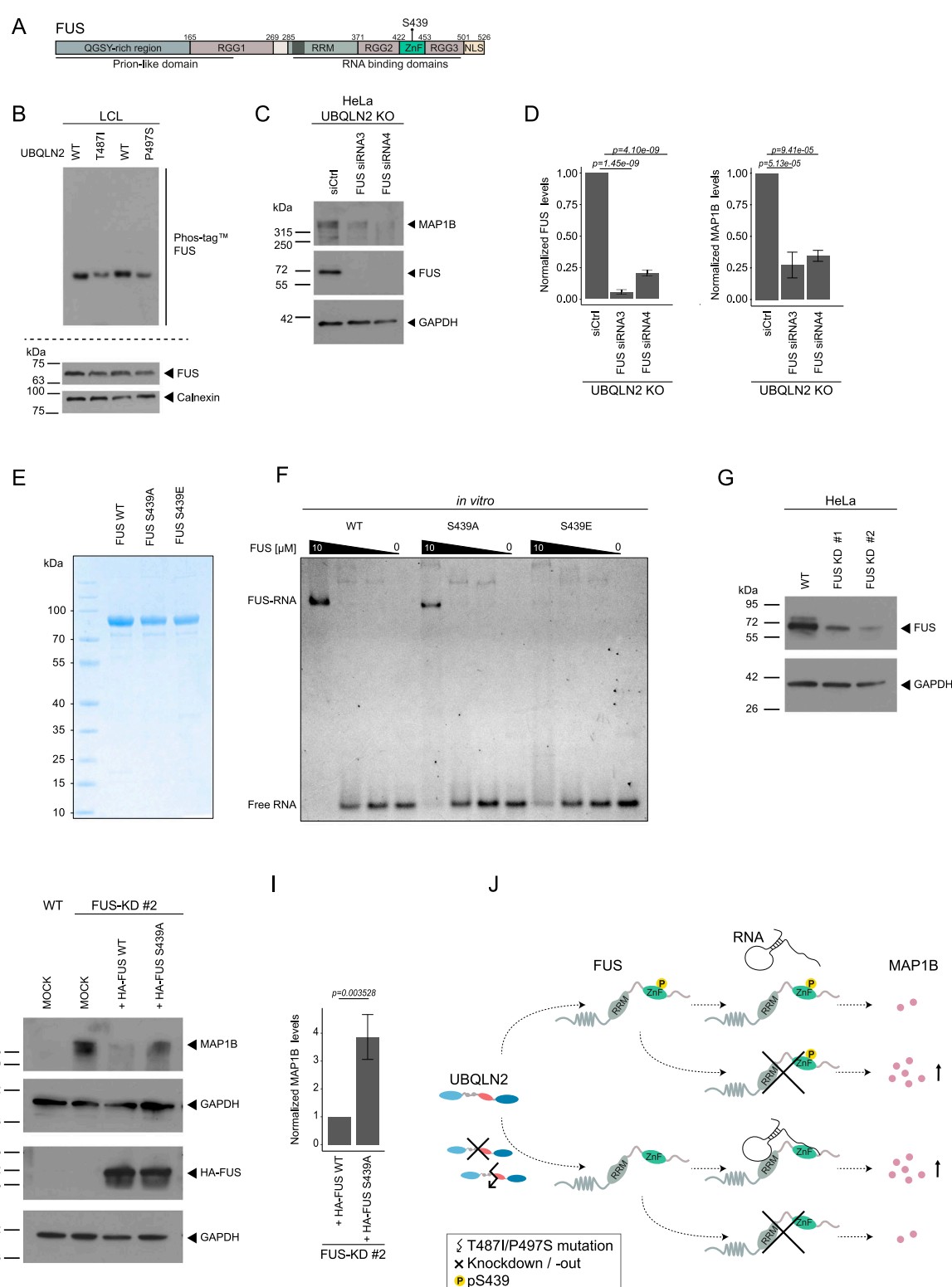

**Figure 6. Phosphorylation of FUS S439 modulates FUS-RNA binding and MAP1B protein levels.**
**(A)** Schematic representation of FUS. SYGQ-rich, serine, tyrosine, glycine, glutamine-rich domain; RRM, RNA recognition motif; RGG, arginine-glycine-glycine–rich region; NLS, nuclear localization signal; ZnF, zinc finger domain. **(B)** Immunoblot analysis of the Phos-tag gel– (upper panel) and SDS–PAGE (lower panel)–separated lysates derived from UBQLN2 WT and amyotrophic lateral sclerosis–mutant LCLs. **(C)** Immunoblot of UBQLN2 KO HeLa cells treated with two different siRNAs targeting *FUS* or a nontargeting siRNA control (siCtrl). **(C, D)** Quantification of MAP1B and FUS protein levels from (C). Data represent mean ± SD. Statistical analysis (n = 3) of

Based on its versatile functions in neurons, disruption of MAP1B functions can result in severe consequences. Mice and *Drosophila* with defective MAP1B showed a variety of phenotypes, including embryonic or postnatal lethality, neuronal migration alterations, abnormalities in brain structure, and severely affected dendrites and axon development (Edelmann et al, 1996; Gonzalez-Billault et al, 2000; Hummel et al, 2000; Meixner et al, 2000). Conversely, increased levels of MAP1B are found in various human neurological diseases. An intriguing example is fragile X syndrome, a common form of inherited mental retardation which is caused by loss-of-function mutations in the fragile X mental retardation 1 (*FMR1*) gene. In healthy conditions, the fragile X mental retardation protein (FMRP) binds to *MAP1B* mRNA and represses its translation, whereas this repression is absent in the disease (Darnell et al, 2001; Lu et al, 2004). Moreover, Allen et al (2005) reported increased levels of MAP1B-LC1 in giant axonal neuropathy, a devastating sensory and motor neuropathy, caused by impaired proteasomal degradation. They observed cell death in neurons overexpressing MAP1B (Allen et al, 2005). In addition, Parkinson disease protein 7 (Park7/DJ-1) was shown to act as a chaperone that inhibits MAP1B aggregation. Loss-of-function mutations in DJ-1 contribute to the pathogenesis of early-onset Parkinsonism and enhance MAP1B-LC1 aggregation. In this study, MAP1B-LC1 overexpression also led to endoplasmic reticulum stress-induced apoptosis (Wang et al, 2011). Recently, it was suggested that loss of microtubule stability in patients of spinal muscle atrophy (SMA) is a result of MAP1B up-regulation, causing tubulin tyrosination (Bora et al, 2021). Furthermore, an ALS-associated mutation in FUS (R521C) was observed to affect the translation regulator function of FMRP at the synapse, leading to an increased expression of *MAP1B* mRNA (Blokhuis et al, 2016).

In the search for a link between UBQLN2 and MAP1B, we detected a significant down-regulation of the phosphosite pS439 in FUS in ALS patient cells carrying UBQLN2 mutations. FUS is mainly localized in the nucleus but shuttles back and forth between nucleus and cytoplasm (Zinszner et al, 1997; Ederle et al, 2018). Several studies described the cytoplasmic accumulation of FUS in ALS patients and cell and mouse models (Dormann & Haass, 2011; Mackenzie & Neumann, 2017). Phosphorylation of FUS was shown to regulate phase separation, aggregation, and toxicity (Deng et al, 2014b; Monahan et al, 2017). However, all of these phosphorylation events occurred in the low-complexity, prion-like domain of FUS. Although S439 FUS has so far not been associated with ALS, this site is especially interesting given its localization in the zinc finger domain of FUS and its seemingly constitutive phosphorylation

status. Our findings in vitro and in cellulo suggest that phosphorylated FUS is prevented from binding to RNA and helps to restrict MAP1B abundance. Hence, in this regard, FUS pS439 might act as a repressor of *MAP1B*.

Evidently, the underlying mechanism awaits full elucidation as it is currently not clear how increased RNA binding of FUS upon pS439 dephosphorylation leads to elevated MAP1B protein levels. A first step toward this would be to test whether the specific RNA binding properties of the different FUS proteoforms are retained in the case of physiologically relevant targets such as *MAP1B* mRNA. Although the FUS' RGG3 domain has been reported to bind the *MAP1B* 5'-UTR G quadruplex sequence and likely directly competes with FMRP (Imperatore et al, 2020), it is not known whether the ZnF domain contributes to the binding of FUS to *MAP1B* mRNA (Lagier-Tourenne et al, 2012). However, the fact that most of FUS-binding sites in *MAP1B* are clustered in the 3'UTR that harbors several GUGGU sequences which are the primary motif known to be recognized by FUS ZnF indicates that this might indeed be the case. Lastly, it is unclear how UBQLN2 ALS mutations result in changes in FUS phosphorylation. Although UBQLN2 is able to interact with FUS and regulates FUS-RNA complex dynamics and stress granules formation (Alexander et al, 2018), it remains elusive whether these processes would have a direct impact on the phosphorylation status of FUS. Taken together, we provide evidence for a derailed UBQLN2-FUS-MAP1B axis, which unexpectedly brings together several of the known ALS pathomechanisms.

## Limitations

A major shortcoming of this study is the usage of nonneuronal cells for our screening approaches. Because damaged motoneurons are a key feature of ALS, the biological significance of our findings in HeLa and LCL cells might be limited. Moreover, omics studies in more advanced cell models such as iPSC-derived motoneurons might have resulted in additional candidates. To overcome at least the former drawback, we validated our findings in primary rat and mouse neurons, thereby supporting the physiological relevance of our findings. Further analysis of the UBQLN2-FUS-MAP1B axis would benefit from molecular modeling and structure prediction to determine the role of the ZnF and its phosphorylation status for RNA binding of FUS. Moreover, the link between UBQLN2 and FUS dephosphorylation requires further investigation. Here, it might make sense to look for kinases and/or phosphatases that are regulated by UBQLN2 and affect FUS pS439.

---

the target protein/control protein ratio was performed using one-way ANOVA followed by Dunnett's post hoc test. **(E)** SDS–PAGE of recombinant FUS variants (WT, S439A, S439E) stained with Coomassie blue. **(F)** Electrophoretic mobility shift assay (EMSA) of MBP-FUS-His$_6$ variants (WT, S439A, S439E) and *SON* pre-mRNA containing the stem loop and a downstream GUU (5 nM) (n = 3). **(G)** Immunoblot of FUS WT and KO HeLa cells. **(H)** FUS WT and KO cells transfected with HA-FUS proteoforms (WT and S439A) or left untreated (MOCK; only Lipofectamine). **(I)** Quantification of MAP1B protein levels upon expression of HA-FUS WT and S439A from (I). Data represent mean ± SD. Statistical analysis (n = 3) of the target protein/control protein ratio was performed using *t* test. **(J)** Working model: in cells with UBQLN2 WT, FUS S439 is constitutively phosphorylated, a state where FUS-RNA binding is impaired. UBQLN2 mutations (P497S and T497I) result in a reduction of pS439 and in elevated MAP1B levels, which are also observed upon UBQLN2 KO. Depending on whether UBQLN2 is functional or defective, KO of FUS has opposing effects on MAP1B levels.

# Materials and Methods

**Table of Reagents.**

| Reagent/Resource | Reference or source | Identifier or Cat. no. |
|---|---|---|
| Experimental models | | |
| HeLa | ATCC | Cat. no. CCL-2 |
| HEK293T | ATCC | Cat. no. CRL-3216 |
| LCL UBQLN2-T487I | This study | |
| LCL UBQLN2-P497S | This study | |
| LCL UBQLN2 WT (control for T487I) | This study | |
| LCL UBQLN2 WT (control for P497S) | This study | |
| Recombinant DNA | | |
| MGC Human UBQLN2 Sequence-Verified cDNA (CloneId: 4543266) | Dharmacon | MHS6278-202831640 |
| ORFeome Collab. Human MAP1B ORF w/Stop Codon | Horizon | OHS5893-202503825 |
| Cas9-expressing plasmid pX330 | Addgene | 42230 |
| pLentiCRISPR-HF1 Puro | Addgene | 110850 |
| pAAV-SEPT | Addgene | 25648 |
| pSpCas9(BB)-2A-Puro (PX459) | Addgene | 62988 |
| Antibodies | | |
| FITC anti-alpha tubulin | Abcam | ab64503 |
| Guinea pig anti-MAP1B-LC1 | SYSY | 410 005 |
| Mouse anti-FUS | Santa Cruz | sc-47711 |
| Rabbit anti-calnexin | Proteintech | 10427-2-AP |
| Mouse anti-GAPDH | Thermo Fisher Scientific | AM4300 |
| Mouse anti-GAPDH | Millipore | CB1001 |
| Mouse anti-UBQLN2 | Sigma-Aldrich | WH0029978M3-100UG |
| Rabbit Anti-Acetyl-$\alpha$-Tubulin (Lys40) | Cell Signaling | 5335 |
| Rabbit anti-calnexin | Abcam | ab22595 |
| Rabbit anti-HA | Cell Signaling | 3724S |
| Rabbit anti-MAP1B | Sigma-Aldrich | HPA022275 |
| Rabbit anti-MAP1B | Proteintech | 21633-1-AP |
| Rabbit anti-UBQLN2 | Cell Signalling | 85509S |
| Anti-Mouse IgG (H+L), HRP Conjugate | Promega | W402B |
| Anti-Rabbit IgG (H+L), HRP Conjugate | Promega | W401B |
| Goat Anti-Guinea Pig IgG Antibody, F(ab') 2, HRP | Sigma-Aldrich | AQ108P |
| Goat anti-Mouse Alexa Fluor 647 | Invitrogen | A21235 |
| Donkey anti-Rabbit Alexa Fluor 647 | Invitrogen | A31573 |
| Goat anti-Rabbit Alexa Fluor 488 | Invitrogen | A11008 |
| Goat anti-Rabbit Alexa Fluor Plus 647 | Invitrogen | A32733 |
| Goat anti-Mouse Alexa Fluor Plus 488 | Invitrogen | A32731 |
| Goat anti-Mouse Alexa Fluor 594 | Invitrogen | A11005 |
| Goat anti-Rabbit Alexa Fluor 594 | Invitrogen | A11012 |
| Oligonucleotides and other sequence-based reagents | | |
| UBQLN2-T487I mutagenesis Primer | This study | GTGGGGGTGCTGGGAATCGCTATAGGC |

| Reagent/Resource | Reference or source | Identifier or Cat. no. |
|---|---|---|
| UBQLN2-P497S mutagenesis Primer | This study | GGCCCAGTCACCTCCATAGGCCCCA |
| FUS S439A mutagenesis Primer | This study | CCTGTGAGAATATGAACTTCGCTTGGAGGAATGAATGCAAC |
| FUS S439E mutagenesis Primer | This study | CCACCTGTGAGAATATGAACTTCGAATGGAGGAATGAATGCAACCAGTG |
| UBQLN2 sgRNA | This study | CACCGCCTACTTCCCTCACTCCCTT |
| UBQLN2 KO sgRNA1 | This study | CACCGTTTCGAATCCCGATCTGATG |
| UBQLN2 KO sgRNA2 | This study | CACCGACGCAGCCTAGCAATGCCGC |
| UBQLN2 KO sgRNA3 | This study | CACCGACCCCCAAACTGCTCTTGTG |
| FUS KO sgRNA1 | This study | CACCGGTGGTTACAACCGCAGCAG |
| FUS KO sgRNA2 | (Wang et al, 2018) | CACCGGAACTCAGTCAACTCCCCA |
| Rat Ubqln2 shRNA | This study | GCTTCAAATCGCAAACCGA |
| ON_Targetplus siRNA FUS | Dharmacon | J-009497-09-0005, J-009497-10-0005 |
| TBP PrimePCR | Bio-Rad | qHsaCID0007122 |
| ACTB PrimePCR | Bio-Rad | qHsaCED0036269 |
| GAPDH PrimePCR | Bio-Rad | qHsaCED0038674 |
| MAP1B PrimePCR | Bio-Rad | qHsaCID0012114 |
| Chemicals, enzymes, and other reagents | | |
| Bafilomycin A1 | Biomol | Cay11038-1 |
| Lipofectamine 2000 | Invitrogen | 11668019 |
| X-tremeGENE HP DNA Transfection Reagent | Merck | 6366236001 |
| Bortezomib | LC Labs | B-1408 |
| DMEM | GIBCO | 61965-026 |
| DMSO | AppliChem | A3672, 0100 |
| Lipofectamine RNAimax | Invitrogen | 13778-150 |
| PhosSTOP | Roche | 4906837001 |
| EDTA-free Protease Inhibitor Cocktail | Roche | 4693132001 |
| Anti-HA agarose | Thermo Fisher Scientific | 88836 |
| HA Peptide | Sigma-Aldrich | I2149 |
| Software | | |
| ImageJ software – Fiji | | version 1.8.0 |
| MaxQuant | | version 1.6.0.1 |
| Perseus | | version 1.6.5.0 |
| Zenblue | Zeiss | version 2.5 |
| DAVID | https://david.ncifcrf.gov/ | version 6.8 |
| String | https://string-db.org/ | version 11.5 |
| Cytoscape | http://www.cytoscape.org | version 3.8.0 |
| Other | | |
| SuperSep Phos-tag (50 $\mu$mol/l), 7.5% | FUJIFILM Wako Chemicals | 198-17981 |
| Quant-iT PicoGreen dsDNA Assay Kit | Life Technologies | P7589 |
| TruSeq Stranded Total RNA Library Prep Kit with Ribo-Zero | Illumina | 20020612 |
| PureLink Genomic DNA Extraction Kit | Invitrogen | K182002 |
| QIAquick PCR purification kit | QIAGEN | 28104 |
| High Pure RNA isolation kit | Roche | 11828665001 |

| Reagent/Resource | Reference or source | Identifier or Cat. no. |
|---|---|---|
| RNeasy Mini Kit | QIAGEN | 74104 |
| Transcriptor First Strand cDNA Synthesis Kit | Roche | 04379012001 |
| qScript XLT 1-Step RT-qPCR ToughMix | Quantabio | 95132 |
| QExactiveHF mass spectrometer | Thermo Fisher Scientific | |
| Easy-nLC1200 | Thermo Fisher Scientific | |
| EasyLC 1000 | Thermo Fisher Scientific | |
| CFX96 qPCR system | Bio-Rad | |
| Agilent 2100 BioAnalyzer | Agilent | |
| Illumina HiSeq4000 | Illumina | |
| Zeiss LSM800 oil immersion 60x objective | Zeiss | |
| AssayMap Bravo | Agilent | |
| Q Exactive (QE) Plus mass spectrometer | Thermo Fisher Scientific | |

## UBQLN2 ALS patient cells

The LCL line with the p.P497S mutation in UBQLN2 was derived from a male patient with the bulbar onset of familial ALS at the age of 54 yr. The patient suffered from FTD comorbidity. Both his mother and sister were affected by the disease and carried the mutation, in agreement with an autosomal dominant mode of inheritance. He died 28 mo after the onset of ALS. Age at the onset of disease of the patient's mother was 79 yr, and she died after another 23 mo. Age at the onset of disease of the patient's sister was 51 yr, and she also died 23 mo after the onset. Both the mother and the sister did not show signs of FTD. The LCL line with the p.T487I mutation was derived from a pre-symptomatic, currently 54-yr-old healthy carrier of this UBQLN2 mutation. Her mother was affected by bulbar onset ALS at 57 yr and died after 27 mo. Moreover, two of her brothers died from ALS and carried the mutation. Age at the onset of disease was 48 and 36 yr in the two brothers, and disease course was 65 and 26 mo, respectively. Both had a spinal onset of disease. Patients were subject to whole-exome sequencing, and genetic variants in other known ALS disease genes were excluded. The two control LCL cell lines were obtained from a 65-yr-old male and a 54-yr-old female healthy individual without ALS and UBQLN2 mutation.

For the collection and use of blood cells from ALS patients and for whole-exome sequencing of blood DNA, written informed consent was obtained from all individuals. The experiments have been approved by the local Ethical Committees of the Medical Faculties Ulm (Ulm University) and Mannheim (ethical committee II of the University of Heidelberg). Approval numbers are Nr. 19/12 and 2020-678N, respectively.

## Gateway cloning and generation of tagged overexpression cell lines

Using gene-specific primers attB-sites were added to ORFs, which were then introduced by BP reaction into the Gateway entry vector pDONR233. ORFs were transferred by LR reaction to the pHAGE-Ntap Gateway destination vector. Stable cell lines were generated by lentiviral transduction with infectious particles produced by 293T cells.

## Generation of endogenously mutated UBQLN2 HeLa cells

Mutations in endogenous *UBQLN2* were introduced according to Kaulich and Dowdy (Kaulich & Dowdy, 2015). A *UBQLN2*-specific sgRNA was designed using the Broad Institute GPP gRNA designer tool and cloned into pX330. The right and left homology arms were amplified from genomic DNA of HeLa cells and cloned into pAAV-SEPT. Site-directed mutagenesis was performed to introduce UBQLN2-T487I and UBQLN2-P497S mutations. HeLa cells were transfected with pX330 and pAAV-SEPT using Lipofectamine 2000 (Life Technologies) according to the manufacturer's instructions. Cells were selected using G418. Correct introduction of the mutations was verified on genomic DNA (PureLink Genomic DNA Extraction Kit; Invitrogen) by PCR and subsequent sequencing.

## Generation of KO cell lines

For the generation of UBQLN2 KO lines, sgRNAs against *UBQLN2* were designed and cloned into the pLentiCRISPR-HF1 Puro plasmid (#110850; Addgene). Infectious particles produced by 293T cells were used to generate stable KO cell lines via lentiviral transduction. A single-cell clone was used for the experiments. sgRNAs against *FUS* were cloned into pSpCas9(BB)-2A-Puro (PX459) (#62988; Addgene) and transected with X-tremeGENE according to the manufacturer's protocol.

## Site-directed mutagenesis

The QuickChange Primer Design software (Agilent) was used to design mutagenesis primers. KOD Hot Start Polymerase (Merck Millipore) was used according to the manufacturer's protocol. Forward and reverse primers were first used in individual PCRs before combining PCR mixtures to generate the final mutated PCR

product. The PCR product was purified with the QIAquick PCR purification kit (QIAGEN) and amplified in *Escherichia coli*.

## Primary rodent neurons with reduced UBQLN2 levels

An shRNA-targeting rat *Ubqln2* (gatccccGCTTCAAATCGCAAACCGAtt-caagaga-TCGGTTTGCGATTTGAAGCtttttggaaa) was cloned downstream of an H1 promoter into a FU3a vector which co-expresses TagRFP from a human ubiquitin promoter. Infectious lentiviral particles were packaged in 293FT cells as described (Guo et al, 2018). Primary rat hippocampal and cortical neurons were prepared from embryonic day-19 rats as described before (Guo et al, 2018). After 3 d, primary neurons were transduced with lentivirus and either fixed in PFA for immunofluorescence or harvested in 1× loading buffer 5 d later. Mouse hippocampal and cortical neurons were isolated from embryonic day-16.5 mice as previously described (Harbauer et al, 2022). On day one in vitro, cells were transfected with either control shRNA or a shRNA targeting Ubqln2. All animal experiments were performed in accordance with the relevant guidelines and regulations of the Government of Upper Bavaria.

## Transfections

For RNAi, cells were transfected with Lipofectamine RNAimax (Invitrogen) and ON-target individual siRNAs (Dharmacon) according to standard protocols. Cells were harvested 72 h after transfection. For transfection of plasmids, Lipofectamine 2000 (Invitrogen) was used according to the manufacturer's instructions.

## Immunoblotting

After washing with DPBS (GIBCO), cells were either directly boiled (95°C) in sample buffer (200 mM Tris–HCL, 6% SDS, 20% glycerol, 300 mM DTT and bromophenol blue) or were lysed in RIPA buffer (50 mM Tris, 150 mM NaCl, 0.1% SDS, 0.5% sodium deoxycholate, 1% NP-40, 1× protease inhibitors [Roche], and 1× PhosStop [Roche]). Protein samples were loaded on an SDS–PAGE gel and transferred on a nitrocellulose or PVDF membrane (2 h 15 min, 0.3A). Immunoblots were blocked in 5% milk in TBS-T (TBS with 0.1% Tween20) or in I-Block (Invitrogen). Membranes were incubated overnight at 4°C with the primary antibody. After three washing steps with TBS-T, secondary HRP-coupled antibodies were added for 1 h, and then membranes were washed again with TBS-T. Finally, membranes were covered with ECL solution (PerkinElmer), and chemiluminescence was measured. Immunoblot quantification was performed in ImageJ using the "Gel Analysis" tool for measurement of the blot band density.

## Phos-tag assay

SuperSep Phos-tag precast gels (7.5%; Wako Chemicals) were used to detect phosphorylation changes according to the manufacturer's protocol. In brief, cell pellets were lysed in urea buffer (9 M Urea, 50 mM Tris, pH 8, 150 mM NaCl, 1x protease inhibitors [Roche], 1× phosphatase inhibitor [Roche]), and protein concentration was adapted. Then proteins were precipitated with TCA (final volume 20%) to remove impurities, and afterward, TCA was washed off with acetone. Protein precipitates were solved in sample buffer (200 mM

Tris–HCL, 6% SDS, 20% glycerol, 5% 2-mercaptoethanol, and bromophenol blue). After electrophoresis, gels were 3× washed in transfer buffer (25 mM Tris, 192 mM glycine, 20%, [vol/vol] methanol) containing 10 mM EDTA for 10 min and once immersed in transfer buffer without EDTA. Gels were transferred on a PVDF membrane and further processed as conventional immunoblots.

## Immunofluorescence

Primary hippocampal rodent neurons and HeLa cells were grown on coverslips and washed three times with DPBS (GIBCO) before fixation. LCL cells were first washed with DPBS and then plated on lysine-coated coverslips for 1 h before fixation. Fixation was according to cell type for 10–15 min with 4% PFA or for 10 min with ice-cold methanol. Cells were incubated for 10 min in 0.5% Triton-X (Merck) for permeabilization and blocked in 1% BSA for either 1 h at room temperature or at 4°C overnight. Primary and secondary antibodies were each incubated for 1 h at room temperature in the dark. Mouse neurons were incubated in primary antibodies overnight at 4°C and in secondary antibodies for 2 h at room temperature in the dark. For acetylated-tubulin staining, fixed HeLa cells were stained overnight at 4°C with the primary antibody solved in 1% BSA, 10% goat serum, and 0.2% Triton-X. The $\alpha$-tubulin-FITC antibody and secondary antibodies were incubated for one. Coverslips of mouse neurons were mounted with Fluoromount-G (Invitrogen) and imaged with a spinning disk confocal on a Nikon Ti2 microscope, using a Plan Apo Lambda 60× oil immersion objective. Coverslips were mounted with Prolong Gold with DAPI (Invitrogen) on frosted microscope slides (Thermo Fisher Scientific) and imaged with a Zeiss LSM800 confocal microscope with an oil immersion objective (63× or 40×). Images were analyzed with ZEN blue (Zeiss, version 3.4) and ImageJ. For quantification of immunofluorescence images, the mean gray value of in total 100 cells from 5 coverslips (20 cells of one cover slip from four different cover slip regions are considered as one biological replicate) were analyzed per condition with ImageJ.

## qRT-PCR

Total RNA from LCL or HeLa cells was extracted with the High Pure RNA isolation kit (Roche) and reverse-transcribed into cDNA using the Transcriptor First Strand cDNA Synthesis Kit (Roche). Real-time quantitative PCR was performed in a CFX96 qPCR system (Bio-Rad) using PrimePCR PCR primers (Bio-Rad). Relative target gene mRNA expression was calculated against the expression of the reference genes *ACTB*, *TBP*, and *GAPDH* with the CFX Maestro Software (Bio-Rad, version 2.1). RNA extraction from mouse neurons was performed using the RNeasy Mini Kit (QIAGEN), and reverse transcription of cDNA was performed with qScript XLT 1-Step RT-qPCR ToughMix (Quantabio). qRT-PCR was conducted in a MIC real-time PCR cycler (Bio Molecular Systems), and relative target gene expression was normalized to GAPDH.

## Electrophoretic mobility shift assays

The stem loop RNA from the *SON* transcript gene (GGAU-CUUUAACUACUCAAGAUACUGAACAUGACAUGGUA) was chemically synthesized with the addition of a Cy5-label (metabion). Then 5 nM

of Cy5-labeled RNA was mixed with varying amounts of FUS WT, S439A, and S439E (0–10 $\mu$M). Binding reactions (20 $\mu$l) were incubated in binding buffer (20 mM NaPO4 [pH8], 150 mM NaCl, 5% glycerol, 1 mM DTT, 5 mM MgCl$_2$, 0.5 mg/ml BSA, 0.1 mg/ml yeast tRNA, and 1 U/$\mu$l RNase inhibitor [Thermo Fisher Scientific]) for 20 min at RT before loading onto a 1-mm thick nondenaturing polyacrylamide gel (6%) in 0.5× TBE. Gels were run at 100 V for 40 min at RT. Gels were imaged with a Bio-Rad ChemiDoc MP Imaging system.

## Recombinant protein expression and purification

Expression and purification of recombinant MBP-FUS-His$_6$ proteins were performed as described before (Hofweber et al, 2018). Briefly, the respective bacterial expression vectors (WT, S439A, and S439E) were transformed into *E. coli* BL21-DE3-Rosetta LysS and grown in standard lysogeny broth (LB) medium. At an OD (600 nm) of ~0.8, cells were induced with 0.1 mM IPTG for 22 h at 12°C. Cells were lysed in resuspension buffer (50 mM NaPO$_4$ [pH 8.0], 300 mM NaCl, 10 $\mu$M ZnCl$_2$, 40 mM imidazole, 4 mM $\beta$ME, 10% glycerol), and tandem-affinity purification using Ni-NTA Agarose (QIAGEN) and amylose resin (NEB) was performed. The protein was washed with resuspension buffer and eluted in resuspension buffer including 250 mM imidazole and 20 mM maltose, respectively, and dialyzed against storage buffer buffer (20 mM NaPO$_4$ [pH 8.0], 150 mM NaCl, 1 mM DTT, 5% glycerol). Concentrations of purified proteins were determined from their absorbance at 280 nm using $\varepsilon$ calculated by ProtParam. The 260/280 nm ratios of purified proteins were between 0.6 and 0.7.

## RNASeq

Library preparation was performed using the TruSeq Stranded Total RNA Library Prep Kit with Ribo-Zero (Illumina). Briefly, RNA was isolated from whole-cell lysates using the High Pure RNA isolation kit (Roche), and the RNA integrity number (RIN) was determined with the Agilent 2100 BioAnalyzer (RNA 6000 Nano Kit; Agilent). For library preparation, 1 $\mu$g of RNA was depleted for cytoplasmatic rRNAs, fragmented, and reverse-transcribed with the Elute, Prime, Fragment Mix. A-tailing, adapter ligation and library enrichment were performed as described in the High Throughput protocol of the TruSeq RNA Sample Prep Guide (Illumina). RNA libraries were assessed for quality and quantity with the Agilent 2100 BioAnalyzer and the Quant-iT PicoGreen dsDNA Assay Kit (Life Technologies). RNA libraries were sequenced as 150-bp paired-end runs on an Illumina HiSeq4000 platform. The STAR aligner (Dobin et al, 2013) (v 2.4.2a) with modified parameter settings (–twopassMode = Basic) is used for split-read alignment against the human genome assembly hg19 (GRCh37) and UCSC known gene annotation. To quantify the number of reads mapping to annotated genes, we used HTseq-count (Anders et al, 2015) (v0.6.0). Fragments per kilobase of transcript per million fragments mapped (FPKM) values are calculated using custom scripts. Differential expression analysis was performed using the R Bioconductor package DESeq2 (Love et al, 2014).

## Whole-proteome analysis

Cell pellets were resuspended in urea buffer (9 M urea, 50 mM Tris, pH 8, 150 mM NaCl, 1× protease inhibitors [Roche], and 50 $\mu$M PR-619) and

sonicated. The sample was cleared by centrifugation at 2,500$g$, and protein amount was adapted after a BCA assay. DTT (5 mM final) was added and incubated for 25 min at 56°C for protein reduction. Samples were then incubated for 30 min at room temperature in the dark with IAA (14 mM final) for protein alkylation. The reaction was quenched by addition of DTT (5 mM final). The protein mixtures were diluted 1:5 with 1 M Tris–HCl, pH 8.2, to lower the urea concentration. The protein was digested with LysC (FUJIFILM, 2 $\mu$l/100 $\mu$g protein) for 3 h followed by trypsin (0.5 $\mu$g/100 $\mu$g protein) at 37°C overnight. Digestion was stopped by the addition of 10% TFA. Peptide samples were fractioned by C18-SCX custom-made stage tips as described elsewhere (Rappsilber et al, 2007). Briefly, the sample was loaded on a preconditioned stage tip containing 2× SCX disks and 2× C18 disks. Stepwise elution was performed with reversed-phase ion-exchange buffers (ReX Buffer: 0.5% AcOH, 20% ACN) with increasing NH$_4$AcO concentrations (20–500 mM). The collected fractions were desalted on custom-made C18 stage tips before MS analysis. In general, four biological replicates were used per cell line, except for UBQLN2-P497S LCLs and UBQLN2-T487I HeLa, where one replicate was lost during processing. Eluted peptides were loaded onto custom-made 75 mm × 15 cm fused silica capillaries filled with C18AQ resin (Reprosil-Pur 120, 1.9 $\mu$m, Dr. Maisch HPLC) using an Easy-nLC1200 liquid chromatography. Peptide mixtures were separated using an ACN gradient (5–95%) in 0.5% acetic acid on a Q Exactive HF mass spectrometer (Thermo Fisher Scientific). For APEX2 experiments, a 35-min–long gradient was used, whereas a 75-min–long gradient was used for whole proteomics. MS raw data were processed with MaxQuant (version 1.6.0.1) and then analyzed in Perseus (version 1.6.5.0). Functional annotations were performed on the platform DAVID (https://david.ncifcrf.gov/).

## Immunoprecipitation

PBS-washed cell pellets were lysed in MCLB buffer (50 mM Tris, pH 7.4, 150 mM NaCl, 0.5% NP40, and 1× protease inhibitors [Roche]). Lysates were cleared by centrifugation (18,400$g$, 10 min, 4°C), concentrations between samples were adapted, and then samples were filtered through a 45-$\mu$m spin filter (Millipore). Lysates were immunoprecipitated overnight at 4°C with pre-equilibrated anti-HA-agarose (Sigma-Aldrich). Agarose beads were washed 5× with ice-cold MCLB followed by five washing steps with PBS. Afterward, proteins were eluted with HA-Peptide (Sigma-Aldrich). TCA was added to the samples for a final concentration of 20% to precipitate proteins. For TCA removal, precipitated proteins were washed 3× with ice-cold acetone. Precipitated proteins were solved in ammonium bicarbonate (ABC) with 10% acetonitrile, and trypsin was added. After 4 h at 37°C, the tryptic digest was stopped, and samples were desalted on custom-made C18 stage tips before MS analysis.

## Phosphoproteomic sample preparation and analysis

LCL cells were cultured in lysine- and arginine-free DMEM (GIBCO) supplemented with 2 mM glutamine, 10% dialyzed FBS, and antibiotics as well as with 146 mg/ml light (K0; Sigma-Aldrich) or heavy

lysine (K8; Cambridge Isotope Laboratories) and 84 mg/ml light (R0; Sigma-Aldrich) or heavy (R10; Cambridge Isotope Laboratories) arginine, respectively. Cell pellets were lysed in 8 M urea in 50 mM Tris–HCl, pH 8.0. Protein concentrations were determined by a BCA assay, and protein amounts were adjusted to equal concentrations. Cell lysates were reduced by 1 mM DTT, alkylated by 5 mM iodoacetamide, and digested by Lys-C for 4 h. The urea solution was diluted to 1 M urea before tryptic digestion overnight. Resulting peptides were purified and fractionated as described previously (Hu et al, 2019). Briefly, peptides were purified by SPE using HR-X columns in combination with C18 cartridges. Purified peptides were frozen, lyophilized, and fractionated by HpH reversed-phase chromatography (Batth et al, 2014). A total of 96 fractions were mixed with an interval of 12 to yield 12 final fractions. Fractions were acidified, frozen in liquid nitrogen, and lyophilized before phosphopeptides enrichment. Phosphopeptide enrichment was performed on an Automated Liquid Handling Platform (AssayMap Bravo; Agilent) (Post et al, 2017). Fe (III)-NTA cartridges (5 μl) were primed with 0.1% TFA in acetonitrile and equilibrated with loading buffer (0.1% TFA in 80% acetonitrile). Peptides were resuspended in 200 μl loading buffer and loaded onto the cartridges with a flow rate of 5 μl/min. Cartridges were washed twice with 200 μl loading buffer with a flow rate of 10 μl/min. Phosphopeptides were eluted with 100 μl of 1% ammonia in 80% acetonitrile with a flow rate of 5 μl/min. Eluates were acidified with 5 μl of formic acid. Samples were lyophilized and resuspended in 20 μl of 0.1% formic acid for LC–MS/MS analysis. The tip flow-through was desalted by STAGE tips for nonphosphopeptide analysis. LC–MS/MS measurements were performed on a QExactive (QE) Plus and HF-X mass spectrometer coupled to an EasyLC 1000 and EasyLC 1200 nanoflow-HPLC, respectively (all Thermo Fisher Scientific). Peptides were fractionated on a fused silica HPLC-column tip (I.D. 75 μm, New Objective, self-packed with ReproSil-Pur 120 C18-AQ, 1.9 μm [Dr. Maisch] to a length of 20 cm) using a gradient of A (0.1% formic acid in water) and B (0.1% formic acid in 80% acetonitrile in water): samples were loaded with 0% B and separated by 5–30% B within 85 min with a flow rate of 250 nl/min. Spray voltage was set to 2.3 kV and the ion-transfer tube temperature to 250°C; no sheath and auxiliary gas were used. Mass spectrometers were operated in the data-dependent mode; after each MS scan (mass range m/z = 370–1,750; resolution: 70,000 for QE Plus and 120,000 for HF-X), a maximum of 10 or 12 MS/MS scans were performed using a normalized collision energy of 25%, a target value of 1,000 (QE Plus)/5,000 (HF-X), and a resolution of 17,500 for QE Plus and 30,000 for HF-X. MS raw files were analyzed using MaxQuant (version 1.6.2.10) (Cox & Mann, 2008) using a Uniprot full-length homo sapiens database (March 2016), and common contaminants such as keratins and enzymes used for in-gel digestion as the reference. Carbamidomethylcysteine was set as fixed modification, and protein amino-terminal acetylation; serine, threonine, and tyrosine phosphorylation; and oxidation of methionine were set as variable modifications. The MS/MS tolerance was set to 20 ppm, and three missed cleavages were allowed using trypsin/P as enzyme specificity. Peptide, site, and protein FDR based on a forward-reverse database were set to 0.01, minimum peptide length was set to 7, the minimum score for modified peptides was 40, and the minimum number of peptides for identification of proteins was set to one, which must be unique. The "match-between-run" option was used with a time window of 0.7 min. MaxQuant results were analyzed using Perseus (Tyanova et al, 2016).

## Data Availability

The mass spectrometry proteomics data have been deposited to the ProteomeXchange Consortium via the PRIDE partner repository with the data set identifiers PXD029730 for the phosphoproteome analysis of UBQLN2 mutant cells and PXD029747 for the whole-proteome and MAP1B interactome analysis.

## Supplementary Information

## Acknowledgments

We thank all members of the Behrends and Edbauer labs for readily sharing reagents, advice, and critical discussions. This work was supported by the Deutsche Forschungsgemeinschaft (DFG, German Research Foundation) within the frameworks of the Munich Cluster for Systems Neurology (EXC 2145 SyNergy – ID 390857198) and the Collaborative Research Center 1177 (CRC1177 – ID 259130777).

### Author Contributions

L Strohm: conceptualization, data curation, validation, investigation, visualization, methodology, and writing—original draft, review, and editing.
Z Hu: data curation, investigation, and methodology.
Y Suk: investigation.
A Rühmkorf: investigation.
E Sternburg: data curation.
V Gattringer: investigation.
H Riemenschneider: investigation.
R Berutti: data curation, investigation, and methodology.
E Graf: data curation, investigation, and methodology.
JH Weishaupt: resources.
MS Leischner-Brill: investigation and visualization.
AB Harbauer: resources, supervision, and investigation.
D Dormann: resources and validation.
J Dengjel: resources, investigation, and methodology.
D Edbauer: resources and methodology.
C Behrends: conceptualization, resources, data curation, supervision, funding acquisition, validation, visualization, project administration, and writing—original draft, review, and editing.

### Conflict of Interest Statement

The authors declare that they have no conflict of interest.

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
