## [Reviewer comments · Life Science Alliance]

Life Science Alliance

Multi-omics profiling identifies a deregulated FUS-MAP1B axis in ALS/FTD-associated UBQLN2 mutants

Laura Strohm, Zehan Hu, Yongwon Suk, Alina Rühmkorf, Erin Sternburg, Vanessa Gattringer, Henrick Riemenschneider, Riccardo Berutti, Elisabeth Graf, Jochen Weishaupt, Monika Leischner-Brill, Angelika Harbauer, Dorothee Dormann, Jörn Dengjel, Dieter Edbauer, and Christian Behrends

DOI: <https://doi.org/10.26508/lsa.202101327>

Corresponding author(s): *Christian Behrends, Ludwig-Maximilians-University Munich*

Review Timeline:

Submission Date:	2021-12-02
Editorial Decision:	2022-01-17
Revision Received:	2022-05-11
Editorial Decision:	2022-06-06
Revision Received:	2022-06-10
Accepted:	2022-06-14

Transaction Report:

January 17, 2022

Re: Life Science Alliance manuscript #LSA-2021-01327-T

Prof. Christian Behrends
Ludwig-Maximilians-University Munich
Munich Cluster for Systems Neurology
Feodor-Lynen Strasse 17
Munich, Bayern 81377
Germany

Dear Dr. Behrends,

Thank you for submitting your manuscript entitled "Multi-omics profiling identifies a deregulated FUS-MAP1B axis in ALS/FTD-associated UBQLN2 mutants" to Life Science Alliance. The manuscript was assessed by expert reviewers, whose comments are appended to this letter. We, thus, encourage you to submit a revised version of the manuscript back to LSA that responds to all of the reviewers' points.

Thank you for this interesting contribution to Life Science Alliance. We are looking forward to receiving your revised manuscript.

Sincerely,

B. MANUSCRIPT ORGANIZATION AND FORMATTING:

Reviewer #1 (Comments to the Authors (Required)):

In their manuscript, Strohm et al. investigate the functional consequences of ALS/FTD-causing mutations in UBQLN2. Using both patient-derived and gene edited HeLa cells, the authors describe the proteome and transcriptome changes. They identify an increase in MAP1B transcript and protein levels, driven by loss of UBQLN1 activity. Aided by interaction- and phospho-proteomics, they link the MAP1B changes to FUS and describe a UBQLN2-FUS-MAP1B axis that may be perturbed in ALS/FTD.

The manuscript is well written and provides several high-quality data sets that will be a useful resource. The identified UBQLN2-FUS-MAP1B axis will be of high interest to the field. The main points of the manuscript are well supported by the data. However, there are several (relatively minor) points that need to be addressed.

Major points

Multiple testing correction is required for the proteomics data sets.

Minor points

The author provide several major datasets and carry out GO analyses. A bit more interpretation of the GO analyses would be interesting. E.g., for the proteomics data, a possible role in exosomes and the involvement of translation (observed terms translation, ribosome, polyA RNA binding, structural constituent of ribosome) seem striking.

As some replicates were lost for the whole-cell proteomics, please indicate more clearly which experiments have n=3 or n=4.

For the proteomics/transcriptomics data sets, it would be useful to also compare what is similar or different across cells, not just mutants. This could be shown by plotting correlation across hits that are significant in at least one data set.

Figure 3A, please provide scaling of the colors to indicate to what extent genes/protein levels changed.

Figure 4E, there appears to be an error with two protein bands of different sizes being labeled as GAPDH.

Figure 5B, should it be "{greater than or equal to}" instead of "> data points"?

Reviewer #2 (Comments to the Authors (Required)):

In the present work, Strohm et al. presented a combination of different omics screenings (conventional proteomics in combination with a transcriptomics analysis, interactomics, and phosphoproteomics screenings) to understand the impact of ALS-specific mutations of UBQLN2. Briefly, the author observed that UBQLN2 mutations (or reduction of protein levels) lead to an increase of MAP1B (both at the protein and mRNA levels) which may be regulated by FUS. Moreover, the authors also observed a phospho-regulation of MAP1B and FUS mediated by UBQLN2 (a reduction in phosphorylation was detected in both cases in presence of ALS-nutant forms).

In my opinion, this work is an important example of how the combination of different screenings can help to find and focus researchers' attention on new findings that may open new avenues of research in a particular field. Importantly, due to its exploratory nature, this type of experiment could in fact identify interesting mechanisms that could be lost if more conventional hypothesis-driven assays (as the potential connection of phosphorylation of FUS and the levels of MAP1B, highlighted in this manuscript). Moreover, this type of work results in several datasets that can be further evaluated by other researchers, and thus it is also a useful resource for the study of ALS. In this sense, I believe it is an important work that should be published.

However, there are some aspects that need to be addressed/clarified prior to publication:

First of all, at the abstract authors state that "Hence, a deregulated UBQLN2-FUS-MAP1B axis links protein homeostasis, RNA

metabolism, and cytoskeleton dynamics, three molecular pathomechanisms of ALS/FTD.", however, no data was shown demonstrating the biological impact of the alterations identified in this work. To present a statement like this, the authors must establish the causality between the dysregulation UBQLN2-FUS-MAP1B axis and a biological impact (for instance, by evaluating the impact in stabilization and/or alterations of proteolytic events in the MAP1B and FUS KO cells). In my point of view, the results of the authors point to a potential connection between the three proteins, therefore I considered that the author may carefully indicate that the finding in this manuscript points for a potential UBQLN2-FUS-MAP1B axis which needs to be further studied to clarify the exact relationship and mechanism. In line with this, it is not clear what is the importance of the phospho-regulation observed in FUS and MAP1B: in fact, the authors observed that there is an alteration (mainly a downregulation) in some residues (all new residues) but they need to prove that those modifications may have a biological role in this mechanism, for instance, they should do some experiments with phospho-mimetic mutants. This is particularly important in the case of the FUS experiments performed to prove the importance of the FUS protein in this axis; the authors performed some silencing experiments, but no alteration in FUS levels were observed in the mutant cells, thus, in my opinion, the authors should repeat this experiment using constitutive phosphorylated forms of FUS in order to prove that UBQLN2 regulated MAP1B via FUS. Although the results from the phosphoproteomics screenings were relevant for connecting FUS with the regulation of MAP1B levels, it is not clear why the authors decide to do these untargeted experiments to study the UBQLN2-mutants. Which was the evidence for a phospho-dysregulation mediated by mutations in UBQLN2 that may justify such a study? In my opinion, the rationale behind this study is not clear, neither in the abstract nor during the text. And I believe that all the experiments should appear in a natural/justified way. In fact, the authors introduce this experiment by saying that: "MAP1B undergoes extensive phosphorylation, which affects its distribution and function" but there is no evidence for a link between phosphorylation and regulation of MAP1B protein levels, that justifies the untargeted screening.

Other small aspects:

- Although the authors add the raw data to public databases, in my opinion, it will be useful if the authors add the tables with the results from the different screenings as supplementary data.
- Depending on the screening, the authors used different criteria to select the highlighted candidates. In my opinion, the authors should try to establish one criterion and apply always the same.
- Use a corrected p-value (FDR adjusted values) in proteomics analysis as performed for transcriptomics.
- At the volcano plots, instead of highlighting several candidates - which in some cases present different modulations - I think the authors should only highlight the ones they give more attention at the main text.
- Regarding Venn diagrams and the analysis performed to compare between cell types and between proteomics and transcriptomics analysis, the authors should use the % of common instead of the total number of candidates. The comparison of the total number will be biased by the clear difference in the total number of altered proteins/transcripts. Moreover, I believe it will make more sense to first compare the results in the different cell models for the same mutation taking into consideration the tendency (increased and decreased), and then perform the comparison between mutants using the common proteins from the first comparison. Since MAP1B is the unique target consistently altered between all conditions, this will not affect the final result, but I believe it will highlight more interesting candidates (the most stable, therefore most confident candidates).
- It is not clear how the authors determine the values indicated in the text as the average altered proteins.
- A direct comparison between the candidates of the proteomics screening with the candidates from the transcriptomic screening (using up- and down-regulated candidates in separate), will be an interesting analysis to determine whether the alterations observed at the protein levels are due to differences in transcription or degradation of the proteins. This might be an important result, considering the importance of UBQLN2 for the degradative pathways.
- All the western blots analyses should present a respective quantitative evaluation normalized for the proper controls. From what I understood there were conditions where there aren't enough replicates to perform statistical analysis, but in my opinion, the authors should make an effort to have that and present not only the representative images but also a proper quantitative and statistical analysis of the WBs. In the case of figure 5J, the authors should normalize the MAP1B levels to FUS, not GAPDH.

Reviewer #3 (Comments to the Authors (Required)):

Strohm et al. studied the role of ubiquitin-binding protein ubiquilin-2 in the disease context of ALS/FTD. They used two patient-derived lymphoblast and HeLA cells with engineered mutations and performed proteomics as well as interactome and phosphoproteomics to identify a signalling path that might link these mutations to dysregulated homeostasis. Since ALS/FTD are neurodegenerative diseases rather than lymphatic diseases, they confirmed a key finding of their results (upregulation of MAP1B levels) in rat primary neuronal cultures with RNA interference with UBQLN2.

In the course of their studies, they employed phosphoproteomics and identified a critical phosphorylation site at serine S439 that resides in the zinc-finger domain and possibly interferes with RNA binding.

In summary, the authors have put a number of screening assays and confirmatory results that might link some genes and pathways associated with ALS/FTD e.g. UBQLN2 and FUS and a yet unknown possibility that phosphorylation at S439 might be a critical disease-modifying component. The limitations of the study are properly discussed and experiments with neuronal cells after iPSC derived differentiation would be a good way to analyse the pathways in future experiments.

Minor points:

The third sentence in the abstract is unclear and might be revised because it implicates that the mutations in UBQLN2 are

causative to ALS/FTD. Is this really supported by data or is this rather a disease-modifying mutation or are mutations associated? If a causative role has been supported by a single gene mutation, one should expect a phenotype e.g. in CRISPR-Cas generated mouse models or in human derived iPSC generated neuronal phenotypes.

The discussion would benefit from a paragraph that discusses the limitations and uncertainties of the present data set in depth (non-neuronal screening platform, various thresholds in the different omics assays, lack of phosphorylation analysis of FUS-S439 in patient derived neuronal phenotypes etc.) and a clear vision of the next steps needed to show causal links in future experiments.

We would like to thank all the reviewers for their time, effort and their valuable comments. We greatly appreciate the reviewers' feedback and are confident that the changes we made improved our paper substantially.

Reviewer #1 (Comments to the Authors (Required)):

In their manuscript, Strohm et al. investigate the functional consequences of ALS/FTD-causing mutations in UBQLN2. Using both patient-derived and gene edited HeLa cells, the authors describe the proteome and transcriptome changes. They identify an increase in MAP1B transcript and protein levels, driven by loss of UBQLN1 activity. Aided by interaction- and phospho-proteomics, they link the MAP1B changes to FUS and describe a UBQLN2-FUS-MAP1B axis that may be perturbed in ALS/FTD.

The manuscript is well written and provides several high-quality data sets that will be a useful resource. The identified UBQLN2-FUS-MAP1B axis will be of high interest to the field. The main points of the manuscript are well supported by the data. However, there are several (relatively minor) points that need to be addressed.

Major points

Multiple testing correction is required for the proteomics data sets.

As suggested by the reviewer, we have performed multiple testing correction (FDR correction) for the proteomics data sets. Proteins which passed the FDR threshold are now highlighted in darker blue and red color in the volcano plots (revised Figure 1C and 1D). These candidates have subsequently been used for the analysis of commonalities among different UBQLN2 mutants (revised Figure 1F and 1G) and GO Terms (revised Figure 1H).

Minor points

The author provide several major datasets and carry out GO analyses. A bit more interpretation of the GO analyses would be interesting. E.g., for the proteomics data, a possible role in exosomes and the involvement of translation (observed terms translation, ribosome, polyA RNA binding, structural constituent of ribosome) seem striking.

We thank the reviewer for this comment. We agree that the GO analysis received too little attention and added a paragraph in this matter in our discussion (line 328-334).

As some replicates were lost for the whole-cell proteomics, please indicate more clearly which experiments have n=3 or n=4.

As advised, we have now indicated the number of replicates in the volcano plots of each omics experiment (revised Figure 1B, 1C, 2C, 2D, 5D and S6).

For the proteomics/transcriptomics data sets, it would be useful to also compare what is similar or different across cells, not just mutants. This could be shown by plotting correlation across hits that are significant in at least one data set.

We thank the reviewer for this input. We addressed this point in two ways: Firstly, we added comparisons of the overlap between the different cell types for both the proteomic and the transcriptomic dataset (new Figure S2A and S3A). Secondly, we

performed a correlation analysis across hits that are significantly altered in at least one data set. This plot is now shown in the new Figure S3B.

Figure 3A, please provide scaling of the colors to indicate to what extent genes/protein levels changed.

As the reviewer proposed, we included a scaling (based on log₂ fold change) to the revised Figure 3A to indicate to what extent the genes/protein levels are changed.

Figure 4E, there appears to be an error with two protein bands of different sizes being labeled as GAPDH.

We thank the reviewer for pointing this out. We corrected this mistake.

Figure 5B, should it be "{greater than or equal to}" instead of "> data points"?

This typo was also corrected.

Reviewer #2 (Comments to the Authors (Required)):

In the present work, Strohm et al. presented a combination of different omics screenings (conventional proteomics in combination with a transcriptomics analysis, interactomics, and phosphoproteomics screenings) to understand the impact of ALS-specific mutations of UBQLN2. Briefly, the author observed that UBQLN2 mutations (or reduction of protein levels) lead to an increase of MAP1B (both at the protein and mRNA levels) which may be regulated by FUS. Moreover, the authors also observed a phospho-regulation of MAP1B and FUS mediated by UBQLN2 (a reduction in phosphorylation was detected in both cases in presence of ALS-nutant forms).

In my opinion, this work is an important example of how the combination of different screenings can help to find and focus researchers' attention on new findings that may open new avenues of research in a particular field. Importantly, due to its exploratory nature, this type of experiment could in fact identify interesting mechanisms that could be lost if more conventional hypothesis-driven assays (as the potential connection of phosphorylation of FUS and the levels of MAP1B, highlighted in this manuscript). Moreover, this type of work results in several datasets that can be further evaluated by other researchers, and thus it is also a useful resource for the study of ALS. In this sense, I believe it is an important work that should be published. However, there are some aspects that need to be addressed/clarified prior to publication:

First of all, at the abstract authors state that "Hence, a deregulated UBQLN2-FUS-MAP1B axis links protein homeostasis, RNA metabolism, and cytoskeleton dynamics, three molecular pathomechanisms of ALS/FTD.", however, no data was shown demonstrating the biological impact of the alterations identified in this work. To present a statement like this, the authors must establish the causality between the dysregulation UBQLN2-FUS-MAP1B axis and a biological impact (for instance, by evaluating the impact in stabilization and/or alterations of proteolytic events in the MAP1B and FUS KO cells). In my point of view, the results of the authors point to a potential connection between the three proteins, therefore I considered that the author may carefully indicate that the finding in this manuscript points for a potential UBQLN2-FUS-MAP1B axis which needs to be further studied to clarify the exact relationship and mechanism.

As the reviewer suggested, we clarified in the last sentence of our abstract, that '[...], our findings point to a deregulated UBQLN2-FUS-MAP1B axis that may link

[...]. Furthermore, we added a paragraph in the end of our discussion, depicting the limitations of our work and emphasizing the need for further studies. In addition, we now provide new data showing that UBQLN2 deficiency leads to elevated levels of total and acetylated tubulin (shown as new Figure 4H and 4I). Given MAP1B's role in binding microtubules and possibly regulating their stability, these alterations are likely consequences of increased MAP1B protein levels.

In line with this, it is not clear what is the importance of the phospho-regulation observed in FUS and MAP1B: in fact, the authors observed that there is an alteration (mainly a downregulation) in some residues (all new residues) but they need to prove that those modifications may have a biological role in this mechanism, for instance, they should do some experiments with phospho-mimetic mutants. This is particularly important in the case of the FUS experiments performed to prove the importance of the FUS protein in this axis; the authors performed some silencing experiments, but no alteration in FUS levels were observed in the mutant cells, thus, in my opinion, the authors should repeat this experiment using constitutive phosphorylated forms of FUS in order to prove that UBQLN2 regulated MAP1B via FUS.

We addressed the reviewers' request and performed experiments with phospho-ablating (S439A) and phospho-mimicking (S439E) mutants. An electrophoretic mobility shift assay (EMSA) with FUS WT and these variants revealed changes in FUS-RNA binding upon substitution of serine with glutamic acid (new Figure 6E and 6F). Furthermore, we observed that re-expression of FUS S439A, mimicking the unphosphorylated state, in FUS KO cells failed to reduce MAP1B protein levels compared to FUS WT (new Figure 6H and 6I). Thus, we now provide evidence that alterations in S439 phosphorylation have a biological role and are likely involved in the regulation of MAP1B.

Although the results from the phosphoproteomics screenings were relevant for connecting FUS with the regulation of MAP1B levels, it is not clear why the authors decide to do these untargeted experiments to study the UBQLN2-mutants. Which was the evidence for a phospho-dysregulation mediated by mutations in UBQLN2 that may justify such a study? In my opinion, the rationale behind this study is not clear, neither in the abstract nor during the text. And I believe that all the experiments should appear in a natural/justified way. In fact, the authors introduce this experiment by saying that: "MAP1B undergoes extensive phosphorylation, which affects its distribution and function" but there is no evidence for a link between phosphorylation and regulation of MAP1B protein levels, that justifies the untargeted screening.

We apologize if our motivation to perform the global phosphoproteomic analysis was not fully comprehensible. First, MAP1B is a 2,468 amino acid large protein with a molecular weight of at least 270 kDa. This imposes an immense challenge for immunoprecipitations, in particular when high coverage is required. Second, databases list between 100 (Uniport) and 250 (PhosphoSitePlus) phosphosites for MAP1B which makes it impossible to set up targeted proteomics. Third, thanks to advanced automation, phosphoproteomics became routine in our lab with no extra effort and expense compared to conventional IP experiments. Hence, we do not consider our phosphoproteomics approach to be an untargeted experiment. Instead, it is literally the only way to unbiasedly and accurately assess the phosphorylation status of MAP1B given its size and abundant sites. Plus, we gained unforeseeable information on a regulated phosphorylation site in FUS which turned out to be likely relevant for the regulation of MAP1B.

Other small aspects:

- Although the authors add the raw data to public databases, in my opinion, it will be useful if the authors add the tables with the results from the different screenings as supplementary data.

We fully agree. All omics data has been added as supplementary tables (Table 1-6).

- Depending on the screening, the authors used different criteria to select the highlighted candidates. In my opinion, the authors should try to establish one criterion and apply always the same.

We now highlighted all proteomic screening candidates which pass the criterion of an FDR-adjusted p-value (q-value) of 0.05 and a Log2 fold change (FC) ≥ 1 or ≤ -1 in dark blue and dark red, respectively (revised Figure 1C and 1D).

- Use a corrected p-value (FDR adjusted values) in proteomics analysis as performed for transcriptomics.

As requested by the reviewer, we marked the FDR adjusted values in our whole cell proteomics analysis and also used this threshold for further analysis (commonality between UBQLN2 mutations and GO terms) shown in the revised versions of Figure 1F, 1G and 1H.

- At the volcano plots, instead of highlighting several candidates - which in some cases present different modulations - I think the authors should only highlight the ones they give more attention at the main text.

We are very thankful for this suggestion, However, after careful re-evaluation we decided that highlighting candidates which are significantly changed in several datasets (proteomics and transcriptomics) are of high interest for the reader. Therefore, we decided not to change the highlighted candidates.

- Regarding Venn diagrams and the analysis performed to compare between cell types and between proteomics and transcriptomics analysis, the authors should use the % of common instead of the total number of candidates. The comparison of the total number will be biased by the clear difference in the total number of altered proteins/transcripts.

We followed the reviewers' suggestion and presented percentage values (based on proteins/genes found in the respective data sets) instead of absolute values. This modified analysis is now presented in Figure 1F, 1G, 2E and 2F.

Moreover, I believe it will make more sense to first compare the results in the different cell models for the same mutation taking into consideration the tendency (increased and decreased), and then perform the comparison between mutants using the common proteins from the first comparison. Since MAP1B is the unique target consistently altered between all conditions, this will not affect the final result, but I believe it will highlight more interesting candidates (the most stable, therefore most confident candidates).

We thank the reviewer for this suggestion. We performed an analysis comparing different cell models carrying the same UBQLN2 mutation. While the overlap is

quite small (only up to 6 proteins), we agree that this information might be valuable for the reader and added the comparison to the manuscript (new Figure S2 and S3A).

- It is not clear how the authors determine the values indicated in the text as the average altered proteins.

To clarify this issue, we added new sub-figures to Figure 1 and 2, presenting the values on which the averages are based (new Figure 1E and 2D).

- A direct comparison between the candidates of the proteomics screening with the candidates from the transcriptomic screening (using up- and down-regulated candidates in separate), will be an interesting analysis to determine whether the alterations observed at the protein levels are due to differences in transcription or degradation of the proteins. This might be an important result, considering the importance of UBQLN2 for the degradative pathways.

As advised by the reviewer, we determined what percentage of the proteomic changes can be explained by alterations at the transcript level. This analysis is now shown in the new Figure S3C. Candidates that are regulated in the same direction in both screenings are highlighted.

- All the western blots analyses should present a respective quantitative evaluation normalized for the proper controls. From what I understood there were conditions where there aren't enough replicates to perform statistical analysis, but in my opinion, the authors should make an effort to have that and present not only the representative images but also a proper quantitative and statistical analysis of the WBs. In the case of figure 5J, the authors should normalize the MAP1B levels to FUS, not GAPDH.

In general, all experiments were done with at least three replicates. However, we agree that quantitative evaluation of critical immunoblots will certainly strengthen our results. Therefore, we focused on Figure 4C and 4F for which we now provide quantification (added as new Figure 4D and 4G). Moreover, we also quantified the levels of MAP1B and FUS in the former Figure 5I (added as new and revised Figure 6C and 6D). In this particular case, we refrained from normalizing MAP1B protein levels to those of FUS (instead of GAPDH) as the latter is affected by siRNA treatment and would therefore eliminate the effect of reduced MAP1B levels. To support this notion, a correlation analysis of normalized FUS and normalized MAP1B levels unveiled a strong dependency ($R=0.98$ and $p\text{-value}=3.9e-06$) of MAP1B levels on FUS. Additional immunoblot data (shown in the new and revised Figure 6H and 6I) was also quantified.

Reviewer #3 (Comments to the Authors (Required)):

Strohm et al. studied the role of ubiquitin-binding protein ubiquilin-2 in the disease context of ALS/FTD. They used two patient-derived lymphoblast and HeLA cells with engineered mutations and performed proteomics as well as interactome and phosphoproteomics to identify a signalling path that might link these mutations to dysregulated homeostasis. Since ALS/FTD are neurodegenerative diseases rather than lymphatic diseases, they confirmed a key finding of their results (upregulation of MAP1B levels) in rat primary neuronal cultures with RNA interference with UBQLN2. In the course of their studies, they employed phosphoproteomics and identified a critical phosphorylation site at serine S439 that resides in the zinc-finger domain and possibly interferes with RNA binding. In summary, the authors have put a number of screening assays and

confirmatory results that might link some genes and pathways associated with ALS/FTD e.g. UBQLN2 and FUS and a yet unknown possibility that phosphorylation at S439 might be a critical disease-modifying component. The limitations of the study are properly discussed and experiments with neuronal cells after iPSC derived differentiation would be a good way to analyse the pathways in future experiments.

Minor points:

The third sentence in the abstract is unclear and might be revised because it implicates that the mutations in UBQLN2 are causative to ALS/FTD. Is this really supported by data or is this rather a disease-modifying mutation or are mutations associated? If a causative role has been supported by a single gene mutation, one should expect a phenotype e.g. in CRISPR-Cas generated mouse models or in human derived iPSC generated neuronal phenotypes.

We acknowledge the reviewers' concern about the causation of UBQLN2 mutations to ALS/FTD. However, it is widely accepted in the field that single point mutations in UBQLN2 are causative to ALS/FTD as firstly reported by Deng et al. 2011 (<https://doi.org/10.1038/nature10353>). In addition, according to Le et al., mice expressing ALS-FTD linked UBQLN2 mutations have cognitive deficits, shortened lifespans, and develop motor neuron disease (<https://doi.org/10.1073/pnas.1608432113>). To our knowledge, no human derived iPSC line with UBQLN2 mutations is published so far. In addition, the ALS patients (from which our LCL cells were derived) were subjected to whole exome sequencing and genetic variants in other known ALS disease genes were excluded. A section on the clinical findings of these patients is added to the Methods part.

The discussion would benefit from a paragraph that discusses the limitations and uncertainties of the present data set in depth (non-neuronal screening platform, various thresholds in the different omics assays, lack of phosphorylation analysis of FUS-S439 in patient derived neuronal phenotypes etc.) and a clear vision of the next steps needed to show causal links in future experiments.

We thank the reviewer for the suggestions and included a limitations section in our discussion, emphasizing the shortcomings of our study as well as further studies needed to clarify the exact relationship and mechanism we found.

June 6, 2022

RE: Life Science Alliance Manuscript #LSA-2021-01327-TR

Prof. Christian Behrends
Ludwig-Maximilians-University Munich
Munich Cluster for Systems Neurology
Feodor-Lynen Strasse 17
Munich, Bayern 81377
Germany

Dear Dr. Behrends,

Thank you for submitting your revised manuscript entitled "Multi-omics profiling identifies a deregulated FUS-MAP1B axis in ALS/FTD-associated UBQLN2 mutants". We would be happy to publish your paper in Life Science Alliance pending final revisions necessary to meet our formatting guidelines.

- please add the Twitter handle of your host institute/organization as well as your own or/and one of the authors in our system
- please make sure that all the authors that are listed in the manuscript are also entered in our system and the author order and names match in the manuscript and our system
- please mention in the Materials and Methods the approval for patient cell line generation, and any informed consent received to perform the whole exome sequencing on patient material
- for the generation of primary rat neurons, please include a statement indicating approval and who granted that approval

Figure Check:

- some of the scale bars in Figure 4 could be bolder for better visibility

A. FINAL FILES:

B. MANUSCRIPT ORGANIZATION AND FORMATTING:

Sincerely,

Reviewer #1 (Comments to the Authors (Required)):

The authors addressed my comments fully; I support publication. I congratulate the authors on their nice work.

Reviewer #3 (Comments to the Authors (Required)):

The authors have revised their study and included further paragraphs including shortcomings and limitations as well as statistical modifications according to the reviewer's suggestion. I have no further concerns.

June 14, 2022

RE: Life Science Alliance Manuscript #LSA-2021-01327-TRR

Prof. Christian Behrends
Ludwig-Maximilians-University Munich
Munich Cluster for Systems Neurology
Feodor-Lynen Strasse 17
Munich, Bayern 81377
Germany

Dear Dr. Behrends,

Thank you for submitting your Research Article entitled "Multi-omics profiling identifies a deregulated FUS-MAP1B axis in ALS/FTD-associated UBQLN2 mutants". It is a pleasure to let you know that your manuscript is now accepted for publication in Life Science Alliance. Congratulations on this interesting work.

DISTRIBUTION OF MATERIALS:

Again, congratulations on a very nice paper. I hope you found the review process to be constructive and are pleased with how the manuscript was handled editorially. We look forward to future exciting submissions from your lab.

Sincerely,
